# Medial temporal lobe functional connectivity predicts stimulation-induced theta power

E.A. Solomon [1], J.E. Kragel[2], R. Gross[3], B. Lega[4], M.R. Sperling[5], G. Worrell[6], S.A. Sheth[7], K.A. Zaghloul [8], B.C. Jobst[9], J.M. Stein[10], S. Das[10], R. Gorniak[11], C.S. Inman[3], S. Seger[4], D.S. Rizzuto[2] & M.J. Kahana[2]

Focal electrical stimulation of the brain incites a cascade of neural activity that propagates from the stimulated region to both nearby and remote areas, offering the potential to control the activity of brain networks. Understanding how exogenous electrical signals perturb such networks in humans is key to its clinical translation. To investigate this, we applied electrical stimulation to subregions of the medial temporal lobe in 26 neurosurgical patients fitted with indwelling electrodes. Networks of low-frequency (5–13 Hz) spectral coherence predicted stimulation-evoked increases in theta (5–8 Hz) power, particularly when stimulation was applied in or adjacent to white matter. Stimulation tended to decrease power in the high-frequency broadband (HFB; 50–200 Hz) range, and these modulations were correlated with HFB-based networks in a subset of subjects. Our results demonstrate that functional connectivity is predictive of causal changes in the brain, capturing evoked activity across brain regions and frequency bands.

[1] Department of Bioengineering, University of Pennsylvania, Philadelphia, PA 19146, USA. [2] Department of Psychology, University of Pennsylvania, Philadelphia, PA 19146, USA. [3] Department of Neurosurgery, Emory School of Medicine, Atlanta, GA 30322, USA. [4] Department of Neurosurgery, University of Texas Southwestern, Dallas, TX 75390, USA. [5] Department of Neurology, Thomas Jefferson University Hospital, Philadelphia, PA 19107, USA. [6] Department of Neurology, Department of Physiology and Bioengineering, Mayo Clinic, Rochester, MN 55905, USA. [7] Department of Neurosurgery, Baylor College of Medicine, Houston, TX 77030, USA. [8] Surgical Neurology Branch, National Institutes of Health, Bethesda, MD 20814, USA. [9] Department of Neurology, Dartmouth Medical Center, Lebanon, NH 03756, USA. [10] Department of Radiology, Hospital of the University of Pennsylvania, Philadelphia, PA 19104, USA. [11] Department of Radiology, Thomas Jefferson University Hospital, Philadelphia, PA 19107, USA. Correspondence and requests for materials should be addressed to E.A.S. (email: esolo@pennmedicine.upenn.edu) or to M.J.K. (email: kahana@psych.upenn.edu)

Intracranial brain stimulation is increasingly used to study disorders of human behavior and cognition, but very little is known about how these stimulation events affect neural activity. Though several recent studies have demonstrated the ability to modulate human memory with direct electrical stimulation (DES) of the cortex[1–7], none have described the mechanism by which stimulation yields altered cognitive states. However, understanding how the brain responds to these exogenous currents is necessary to ultimately develop therapeutic interventions that rely on DES[8,9].

Specifically, investigators have long asked whether the brain's intrinsic functional or anatomical architecture can predict how mesoscale stimulation events propagate through the brain. Early work focused on inferred connectivity through stimulation-evoked behavior in rodents[10,11]. More recently, Logothetis and colleagues demonstrated that the effects of electrical stimulation propagated through known anatomical connections in the macaque visual system[12,13]. In humans, corticocortical evoked potentials (CCEPs), measured with intracranial electro-encephalography (iEEG), have also been shown to propagate through anatomical and functional connections[14,15], as has the functional magnetic resonance imaging (fMRI) BOLD (blood-oxygen-level dependent) response to stimulation[16]. These studies provide powerful evidence that the effects of stimulation are determined by the connectivity profile of a targeted region. More broadly, renewed interest in the idea of the brain as a controllable network[17–19] raises a testable hypothesis in need of empirical validation: to what extent does a brain's network architecture predict the cascade of physiologic change that accompanies a stimulation event?

In this study, we asked whether the functional connectivity of a stimulated region predicts where we observe changes in neural activity. To expand on prior work that has examined network architecture and stimulation, we adopted a paradigm that (1) focuses on stimulation's effect on low-frequency (theta) power, a cognitively relevant electrophysiological biomarker, and (2) simultaneously considers the structural and functional connectivity of a targeted region. In 26 neurosurgical patients with indwelling electrodes, we stimulated different regions of the medial temporal lobe (MTL) and asked whether low-frequency coherence predicted modulations of theta power in distributed cortical regions. We showed that coherence was mostly predictive of theta modulation when stimulation occurred in or near a white matter tract, but in those cases, stimulation could evoke sustained increases in theta power even in distant regions. With this initial finding in hand, we expanded our paradigm to consider additional measures of functional connectivity and evoked power at higher frequencies. We principally considered the amplitude envelope of the high-frequency broadband (HFB) signal (50–200 Hz)[20], shown to correlate with the resting-state fMRI (rsfMRI) BOLD correlations that are widely used in network neuroscience[20–23]. We demonstrated that while low-frequency coherence accurately predicts increases in low-frequency power, HFB-based networks can explain decreases of HFB power. Taken together, functional connectivity can predict the widespread changes in local spectral power induced by DES of the MTL.

## Results

**Calculating network-mediated activation.** To determine how direct cortical stimulation propagates through brain networks, we collected iEEG data from 26 patients undergoing clinical monitoring for seizures. Subjects rested passively in their hospital bed while we applied bipolar macroelectrode stimulation at varying frequencies (10–200 Hz) and amplitudes (0.25–1.5 mA) to MTL depth electrodes (see online Methods for details). Rectangular stimulation pulses were delivered for 500 ms, followed by a 3-s inter-stimulation interval (Fig. 1a–d). Each subject received at least 240 stimulation events ("trials") at 1–7 distinct sites in MTL gray or white matter (mean 2.7 sites; see Supplementary Table 2 for stimulation locations). During a separate recording session in which no stimulation occurred, for each subject we computed resting networks of low-frequency (5–13 Hz) coherence, motivated by prior literature that shows robust iEEG functional connectivity at low frequencies[24–27]. These networks reflect correlated low-frequency activity between all possible pairs of electrodes in a subject, during a period when subjects are passively waiting for a task to begin (Fig. 2a).

For each stimulation trial, we computed theta power (5–8 Hz) in 900 ms windows before and after each 500 ms stimulation event, and compared the pre- vs. post-stimulation power across all trials with a paired $t$-test (Fig. 1e-g). Next, we used linear regression to correlate the strength of a stimulation site's network connectivity to a recording electrode with the power $t$-statistic at that electrode (Fig. 2a–d). We included absolute distance as a factor in our regression, to only consider how connectivity relates to stimulation beyond the brain's tendency to densely connect nearby regions[28]. The result is a model coefficient that indicates, independent of distance, the degree to which functional connectivity predicts stimulation-induced change in theta power at a recording site. The regression was repeated using permuted connectivity/evoked power relationships to generate a null distribution of model coefficients against which the true coefficient is compared. We refer to the resulting $z$-score as the "network-mediated activation $(\theta)$," or $NMA_\theta$. High $NMA_\theta$ indicates functional network connectivity predicts observable stimulation-related change in theta power at distant sites.

**$NMA_\theta$ is correlated with proximity to white matter.** At a group level of stimulation sites, $NMA_\theta$ was significantly greater than zero (one-sample $t$-test, $t(71) = 4.18$, $P = 8.2 \times 10^{-5}$; Fig. 3a), indicating that stimulation in the MTL tends to evoke network-driven change in theta power in distant regions. However, we noted substantial heterogeneity between stimulation sites, with some showing little or no ability to modulate network-wide theta activity, as reflected by $NMA_\theta$ near zero. To explain this heterogeneity, we hypothesized that as earlier work demonstrated[13,15,29], structural connections (i.e. white matter tracts) may be key to the propagation of electrical stimulation throughout the brain.

To test whether structural connections play a role in stimulation propagation, we asked whether $NMA_\theta$ was correlated with the proximity of a stimulation site to white matter. If these measures are correlated, it would indicate that functional connectivity is predictive of physiology only insofar as white matter tracts are accessible. We binned stimulation sites according to whether they were placed in gray matter ($n = 32$, lower 50th percentile of distances to white matter), near white matter ($n = 33$, upper 50th percentile of distances to white matter), or within white matter ($n = 7$, manually identified by a neuroradiologist; Fig. 3a; see Supplementary Figure 1 for anatomical placement of each white matter target). We found that $NMA_\theta$ was significantly increasing with white matter placement, relative to a permuted distribution (permuted $P < 0.001$; Fig. 3b). The $NMA_\theta$ for gray matter sites was not significantly different than zero (one-sample $t$-test, $t(31) = 1.4$, $P = 0.18$), while $NMA_\theta$ for sites near or in white matter was significant ($P < 0.05$). This relationship holds in a Pearson correlation agnostic to any electrode categorization ($r = 0.33$, $P = 0.005$; Supplementary Figure 3). This finding does not mean gray matter stimulation fails to induce theta activity, but it does

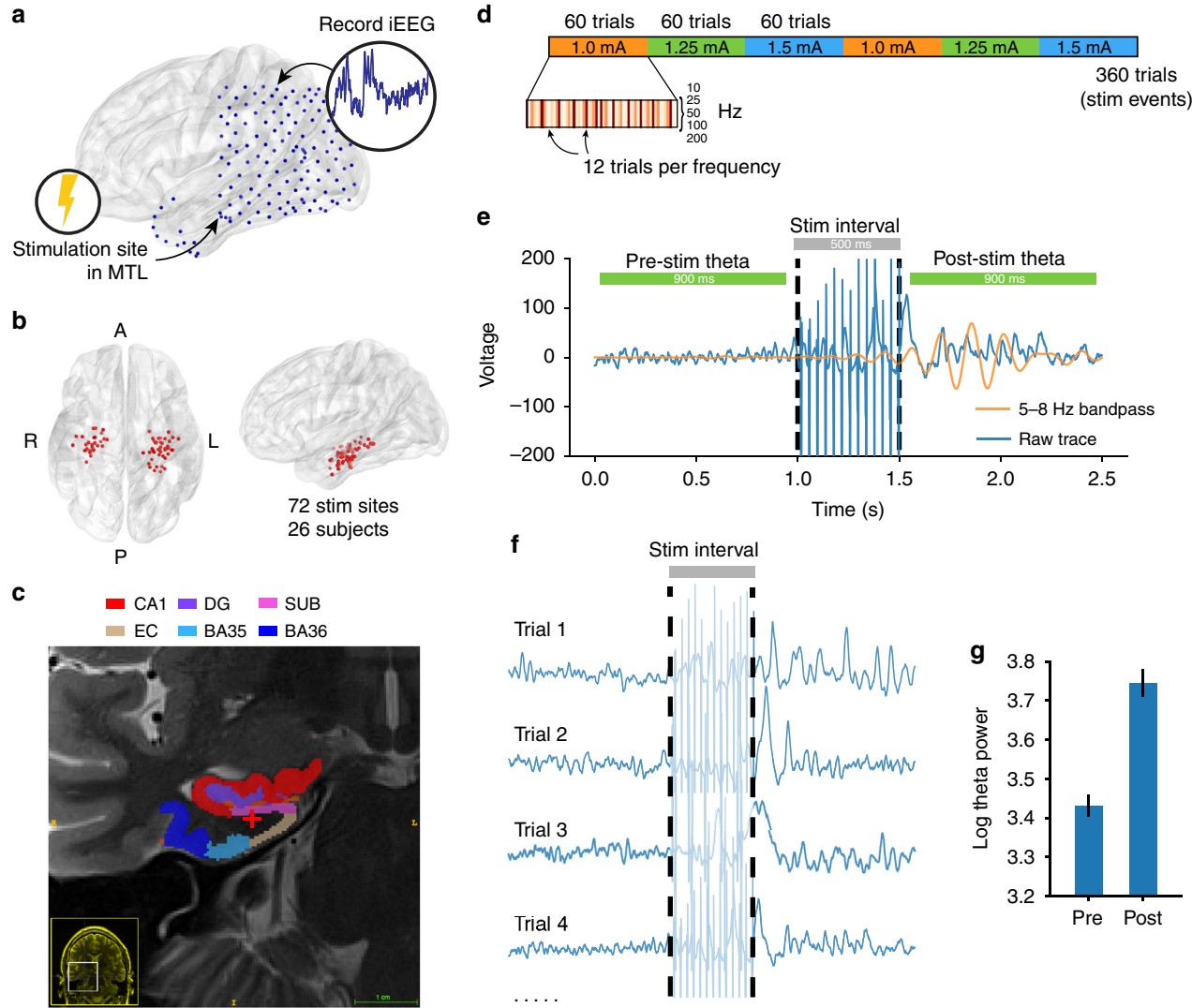

**Fig. 1** Comparison of pre- vs. post-stimulation theta (5–8 Hz) power in an example subject. **a** Each of 26 subjects received a series of 500 ms bipolar stimulation events, at 1–7 sites within the MTL; an example subject schematic is shown here. **b** Anatomical distribution of all MTL stimulation sites in the 26-subject dataset. **c** T2 MRI and MTL subregion segmentation for an example subject. Stimulation location, in white matter, is indicated at the red cross. See Supplementary Figure 1 for subregion labels. **d** Schematic of a typical stimulation session. Each stimulation site receives stimulation at three amplitudes (within the 0.25–2.00 mA range) and five pulse frequencies (50–200 Hz; see Methods for details). During each session, amplitudes are delivered in 60-trial blocks, within which 12 stimulations are delivered at each frequency. For the main results, effects are aggregated across all stimulation parameters; see Supplementary Figure 2 for analysis of stimulation amplitude and frequency. **e** Using the multitaper method, theta power (5–8 Hz) was measured in 900 ms windows preceding and following each stimulation event, with 50 ms buffers before and after stimulation. In an example stimulation event, the 5–8 Hz bandpass signal (orange) is overlaid on the raw bipolar signal (blue), to emphasize a change in pre- vs. post-stimulation theta power. **f** Theta power is extracted in the pre- and post-stimulation intervals for at least 240 events ("trials") per stimulation site. **g** The log-transformed theta power is aggregated for all pre- and post-stimulation intervals separately, for later statistical comparison (Fig. 2)

suggest that stimulation far from white matter tracts may result in theta activity that is uncorrelated with connectivity to remote sites.

To account for the possibility that the theta response is sensitive to the pulse frequency or amplitude of stimulation, we asked whether $NMA_\theta$ differed in accordance with stimulation parameters. Across all stimulation sites, $NMA_\theta$ was marginally greater for trials delivered at a subject's maximum vs. minimum amplitude (paired $t$-test, $t(71) = 1.91$, $P = 0.061$; Supplementary Figure 2A), but no difference was noted across pulse frequencies delivered at 10, 50, and 200 Hz (repeated-measures analysis of

variance, $F = 0.16$, $P = 0.85$; Supplementary Figure 2B). Additionally, raw evoked power was significantly greater for maximum amplitude stimulation ($t(71) = 3.52$, $P = 0.0008$), but did not reliably differ across pulse frequencies ($F = 0.26$, $P = 0.77$; Supplementary Figure 2B). For the remainder of this study, all analyses consider stimulation events aggregated across amplitudes and frequencies.

Taken together, these results show that DES of the MTL can induce spectral power changes across a distributed network of regions, particularly if stimulation occurs in or proximal to white matter. When this occurred, we discovered that functional

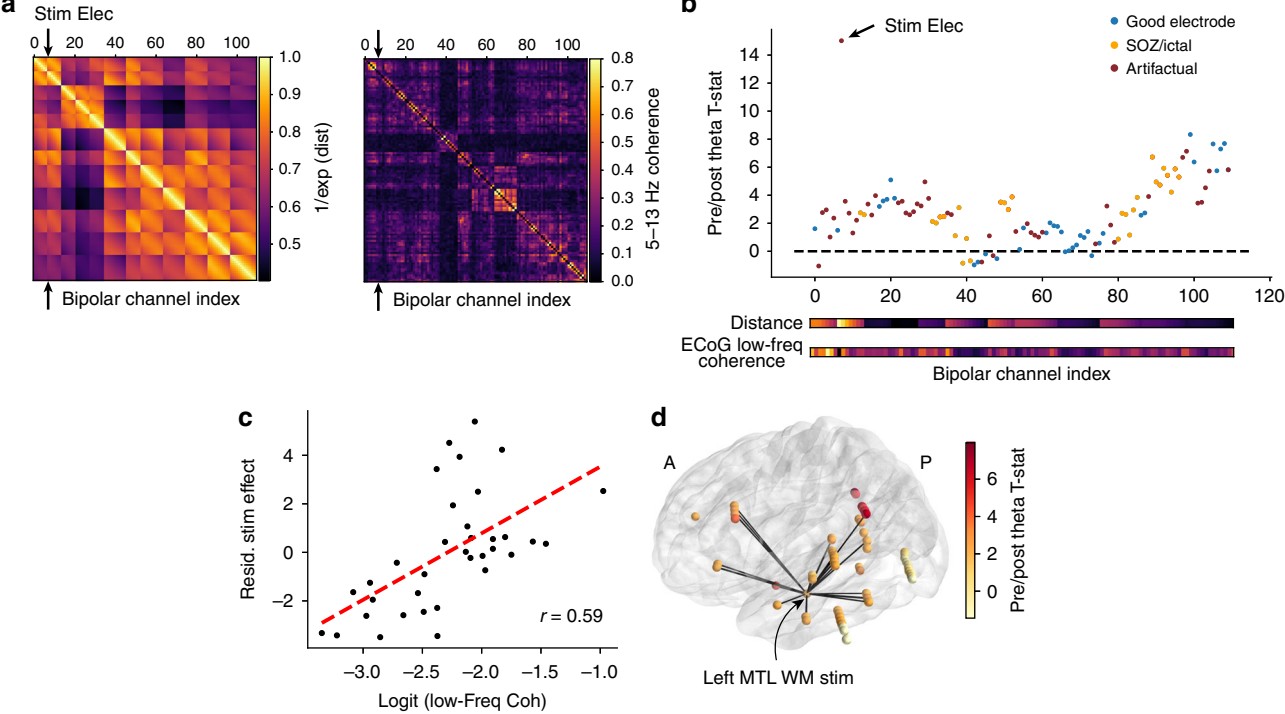

**Fig. 2** Method for determining network-mediated activation (NMA$_\theta$). **a** For each subject, Euclidean distances (left matrix) and functional connectivity (right matrix) are measured for all possible electrode pairs. Distances are linearized as $e^{-(\text{distance})}$, with 1.0 representing no separation between two electrodes. Functional connectivity is the averaged 5–13 Hz multitaper coherence in 1-s windows extracted from a baseline period. **b** Pre- and post-stimulation theta power (Fig. 1e) is compared with a paired $t$-test to generate a $t$-statistic for each electrode. Electrodes are excluded from analysis if they exhibited significant post-stimulation artifact (red, see Methods for details) or were placed in the seizure onset zone or exhibit high inter-ictal spiking (orange). **c** Multiple linear regression is used to correlate the logit-transformed functional connectivity (between a recording electrode and the stimulation electrode) with the power $t$-statistic, independent of distance. To demonstrate this, the distance-residualized $t$-statistic ("stim effect") is plotted against functional connectivity in the example subject. The $z$-scored version of this correlation is referred to as the "network-mediated activation ($\theta$)," or NMA$_\theta$. **d** Rendering of the power $t$-statistic as color on each electrode in the example subject, plotted with the top 10% of functional connections to the stim electrode (black lines)

low-frequency coherence is predictive of where stimulation-related modulations in theta power are observed.

**Network properties of MTL stimulation.** Having shown that stimulation in or near-white matter sites induces distributed changes in theta power, we next sought to characterize the directionality of change. Specifically, high NMA$_\theta$ could be caused by increases in theta power at electrodes with strong functional connectivity to the stimulation target, or decreases in theta power at electrodes with weak connectivity to the stimulation target. To distinguish between these possibilities, we further examined theta power changes among the 16 stimulation sites that exhibited individually significant ($P < 0.05$) NMA$_\theta$ (see Supplementary Table 1 for statistics and anatomical placement of each significant site). In this subset, we measured the average pre- vs. post-stimulation theta power at the five electrodes with the strongest functional connectivity to the stimulation site (controlled for distance), and the five electrodes with the weakest functional connectivity. At strongly connected sites, theta power change was significantly positive (one-sample $t$-test, $t(15) = 5.6$, $P = 4.0 \times 10^{-5}$) and significantly greater than power change at weakly connected sites (paired $t$-test, $t(15) = 6.03$, $P = 1.7 \times 10^{-5}$; Fig. 4b). No significant power change was observed at sites with weak functional connectivity (one-sample $t$-test, $t(15) = 1.5$, $P = 0.15$). Notably, we observed that of the 16 significant sites analyzed here, 15 were placed in or near white matter. We conclude that stimulation causes increased theta power at strongly

connected sites and little to no change in power at weakly connected sites.

Principles of network control theory suggest a relation between the connectivity profile—or network topology—of a stimulation site and the ensuing change in brain activity. Network "hubs," or regions with strong connectivity to the rest of the brain, exert differential effects on overall brain activity vs. non-hubs, or regions with strong connections to only a few areas[18,30]. To directly test whether stimulation propagates differently from hub regions, we asked whether stimulation-induced theta power correlated with the functional "hubness" of a stimulation site. We again took our measure of stimulation-induced activity to be the theta power change at the five recording sites with the strongest functional connectivity to the stimulation site, and tested this metric against the node strength of a stimulation site, an indicator of hubness. For this analysis, we considered all stimulation sites in or near white matter ($n = 40$) as these groups both exhibited significant NMA$_\theta$ (Fig. 3b). When weak hubs (lower tercile of hub scores; $n = 13$) were stimulated, power change at connected recording sites was significantly greater than zero (one-sample $t$-test, $t(12) = 3.6$, $P = 0.003$), but stimulation at strong hubs (upper tercile; $n = 14$) evoked no significant power modulation ($t(13) = 0.15$, $P = 0.87$; Fig. 4d). While counterintuitive, this result could suggest that stimulation at a site with many connections may disperse or blunt the effect of perturbation, yielding lesser activation in downstream regions. Alternatively, hub stimulation does evoke widespread changes in brain activity, but these changes tend to be outside the theta band assessed here.

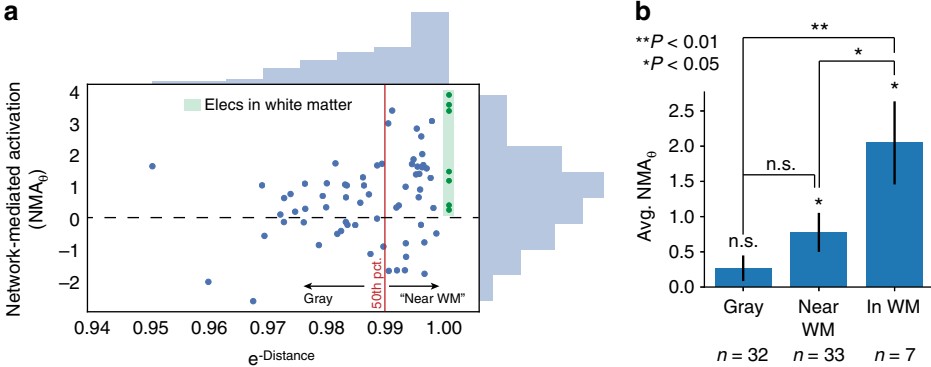

**Fig. 3** Proximity to white matter predicts NMA$_\theta$. **a** Correlation between a stimulation site's distance from nearest white matter with the site's NMA$_\theta$. The 50th percentile of white matter distances divides sites classified as "gray matter" vs. "near white matter." Stimulated contacts in white matter are highlighted in green. See Supplementary Figure 3 for the Pearson correlation of these data ($r = 0.33$, $P = 0.005$). **b** NMA$_\theta$ increases with closeness to white matter, as determined by a permutation test ($P < 0.001$, see Methods) and by noting that NMA$_\theta$ for sites in or near white matter are significantly greater than zero (one-sample $t$-test, $P < 0.05$) while gray matter sites are not ($P = 0.15$). Electrodes placed in white matter have greater NMA$_\theta$ than electrodes near white matter (two-sample $t$-test, $P < 0.05$) or gray matter ($P < 0.01$). Error bars show ±1 SEM; *$P < 0.05$; **$P < 0.01$

Our choice of low-frequency (5–13 Hz) functional connectivity as the basis for predicting distributed changes in theta power was motivated by prior studies that have shown strong, cognitively relevant connectivity at low frequencies, particularly the theta and alpha bands[24,25,27]. However, others have noted significant inter-regional connectivity in the beta and gamma bands[31]. As our study presented a unique opportunity to examine the causal nature of functional connectivity, we asked whether functional connectivity in other frequency bands is also predictive of downstream power modulations. Among all MTL electrodes placed in or near white matter ($n = 40$), we asked whether NMA was significant for networks constructed from any frequency to a maximum of 50 Hz. No frequencies outside the alpha/theta bands exhibited significant group-level NMA$_\theta$, after correction for multiple comparisons (one-sample $t$-test, $P < 0.05$, Benjamini-Hochberg correction; Fig. 4e). This demonstrates that functional networks constructed from high frequencies (>13 Hz) are not predictive of stimulation-induced theta activity.

**Alternative measures of connectivity.** Functional connectivity is a broad domain, generally referring to an array of measures that fundamentally reflect time-series correlations. In addition to the phase-based measures (i.e. spectral coherence) used here, other correlations have also been shown to robustly capture inter-regional functional dynamics in the human brain. Of particular utility in iEEG investigations is the amplitude envelope of HFB (50–200 Hz), shown to reflect neuronal population spiking activity[32] and correlated with fMRI BOLD activation[33]. The slow (<1 Hz) fluctuations of this signal have also been shown to correlate with resting-state functional connectivity (rsfMRI)[23,34,35]. It has recently been demonstrated that stimulation perturbs brain networks in accordance with measures of functional connectedness, including a modulation of remote cortical excitability[14,36].

We therefore sought to determine whether these established measures of intrinsic functional connectivity—HFB amplitude envelope correlation and rsfMRI connectivity—also predicted the location of evoked theta power. To do this, we replicated our procedure for computing NMA$_\theta$, but used HFB amplitude envelope correlation or atlas-based rsfMRI connectivity as predictor variables (see Methods for details; see Fig. 5a for example adjacency matrices). As with the low-frequency coherence networks, the result is a $z$-scored statistic (NMA$_\theta$) that reflects the degree to which a functional network predicts

remote changes in theta power. We emphasize that HFB envelope networks—though based on extracted high-frequency power—are wholly distinct from 30+ Hz coherence networks assessed earlier; the former is a correlation of the slow variation in a power time series, while the latter is a measure of high-frequency phase consistency between two signals.

Though rsfMRI and HFB connectivity measures qualitatively recapitulated our earlier finding—NMA$_\theta$ increases with closeness to white matter—their ability to predict downstream changes in evoked theta power was not significant across all stimulation sites (HFB connectivity, $t(71) = 1.07$, $P = 0.28$; atlas rsfMRI, $t(49) = 0.17$, $P = 0.87$; Fig. 5b-c). We note that slightly fewer stimulation sites were available for the rsfMRI analysis ($n = 50$), due to subjects where atlas-based measures could not be estimated for a sufficient number of electrodes (see Methods for details).

Though HFB envelope and rsfMRI connectivity did not strongly replicate our finding of significant NMA$_\theta$ using low-frequency coherence, several factors could account for this discrepancy. First, stimulation within the unique architecture of the MTL may propagate differently than the cortical surface stimulation used in many prior studies—it is possible that at the cortical surface, HFB/rsfMRI connectivity is better predictive of stimulation effects than low-frequency coherence. Second, different measures of connectivity may differentially predict different kinds of evoked responses. Low-frequency coherence successfully predicts low-frequency power, but may fail to accurately predict modulations at higher frequencies.

**Evoked responses at higher frequencies.** While our choice to examine the effect of stimulation on theta frequencies was theoretically motivated by a vast literature implicating theta oscillations and cognition[37], activity in the HFB range is a useful marker of population neural activity[38], and cognitively relevant oscillatory dynamics are also observed in the alpha, beta, and gamma bands (9–13, 15–25, 30–60 Hz, respectively). To account for the possibility that stimulation evokes activity in these higher frequency bands, we extended our analysis to consider the correlation between low-frequency coherence and induced power in alpha/beta, gamma, and HFB ranges. Furthermore, to address the possibility that HFB-based connectivity networks (Fig. 5a) better predict induced local HFB power, we asked about the correlation between HFB envelope connectivity and induced power across frequency ranges (see Methods for details).

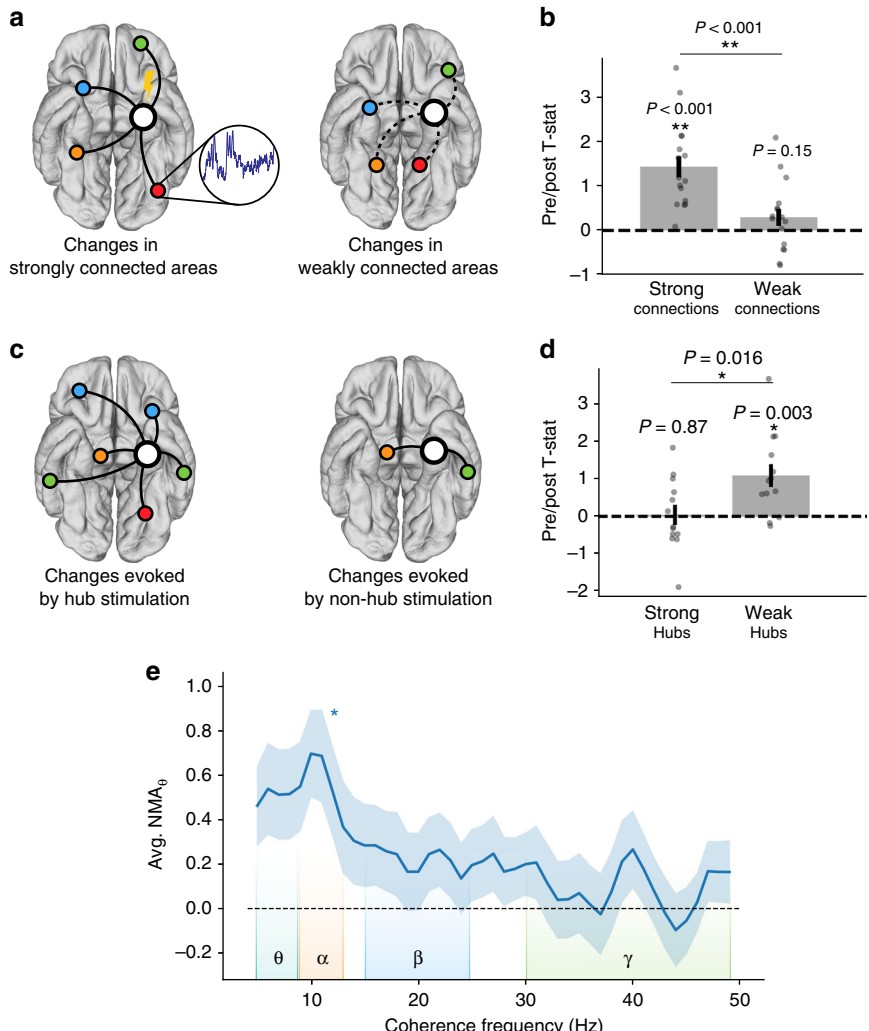

**Fig. 4** Network properties of stimulation-induced theta. **a** Schematic of a stimulation site and its most strongly connected areas (left) or weakly connected areas (right). **b** For each of 16 stimulation sites with significant $NMA_\theta$ ($P < 0.05$), the average post- vs. pre-stimulation theta $t$-statistic is computed for the five strongest-connected electrodes and the five weakest-connected electrodes (controlled for distance). Strongly connected regions are typically areas of the lateral temporal, prefrontal, or inferior parietal cortices. Changes at strongly connected recording sites are significantly greater than changes at weakly connected sites (paired $t$-test, $t(15) = 6.03$, $P = 1.7 \times 10^{-5}$). **c** Schematic of a hub-like stimulation site (left) and a non-hub stimulation site (right). Hub scores are calculated as the node strength, or average of all connection weights to a given electrode. **d** For each of 40 stimulation sites in or near white matter, the average post- vs. pre-stimulation theta $t$-statistic is computed for the five strongest-connected recording electrodes. Stimulation of a weak hub (lower tercile of hub scores, $n = 13$) yields significantly greater change in connected regions than stimulation of a strong hub (upper tercile of hub scores, $n = 14$) (two-sample $t$-test, $P = 0.016$). **e** Average $NMA_\theta$ across all in or near-white matter stimulation sites, as a function of functional connectivity frequency. $NMA_\theta$ is greatest for networks constructed from theta or alpha coherence (5–13 Hz). Corrected for multiple comparisons across all frequencies, $NMA_\theta$ is significantly greater than zero at 11 Hz. Error bars show $\pm 1$ SEM; $*P < 0.05$; $**P < 0.01$

We first assessed whether stimulation evoked any detectable modulation of power in the alpha/beta, gamma, and HFB bands, regardless of relationship to connectivity. To do this, we averaged the pre- vs. post-stimulation $t$-statistic across all electrodes in each subject's brain, for each frequency band. The result is an average $t$-statistic reflecting the stimulation-evoked whole-brain change in power at each frequency band. Across all stimulation sites, stimulation significantly increased power in the theta, alpha/beta, and gamma bands, but significantly decreased power in the HFB range (one-sample $t$-test, false discovery rate (FDR)-corrected $P < 0.05$; Fig. 6a–c). However, the power response to stimulation was not uniform across electrodes within a subject; for electrodes that exhibited a strong theta response, evoked changes were weaker at higher frequencies (Fig. 6d), indicating a theta-specific effect.

Given that stimulation evoked changes in spectral power beyond the theta range, we next asked whether functional connectivity networks predicted these changes (e.g. computing $NMA_{HFB}$). Corrected for multiple comparisons, low-frequency (5–13 Hz) coherence networks only correlated with evoked power in the theta range (one-sample $t$-test, $t(71) = 4.18$, FDR-corrected $P < 0.01$; Fig. 7a). On average, HFB envelope connectivity did not significantly predict power modulation at any frequency band. However, given our earlier finding of decreases in power in the HFB band (Fig. 6), we hypothesized that a null average effect was obscuring heterogenous—but individually significant—responses to stimulation. In other words, for specific subjects, HFB envelope networks could predict increases or decreases in HFB power and yield significant correlations in positive or negative directions. Indeed, HFB functional connectivity significantly predicted HFB

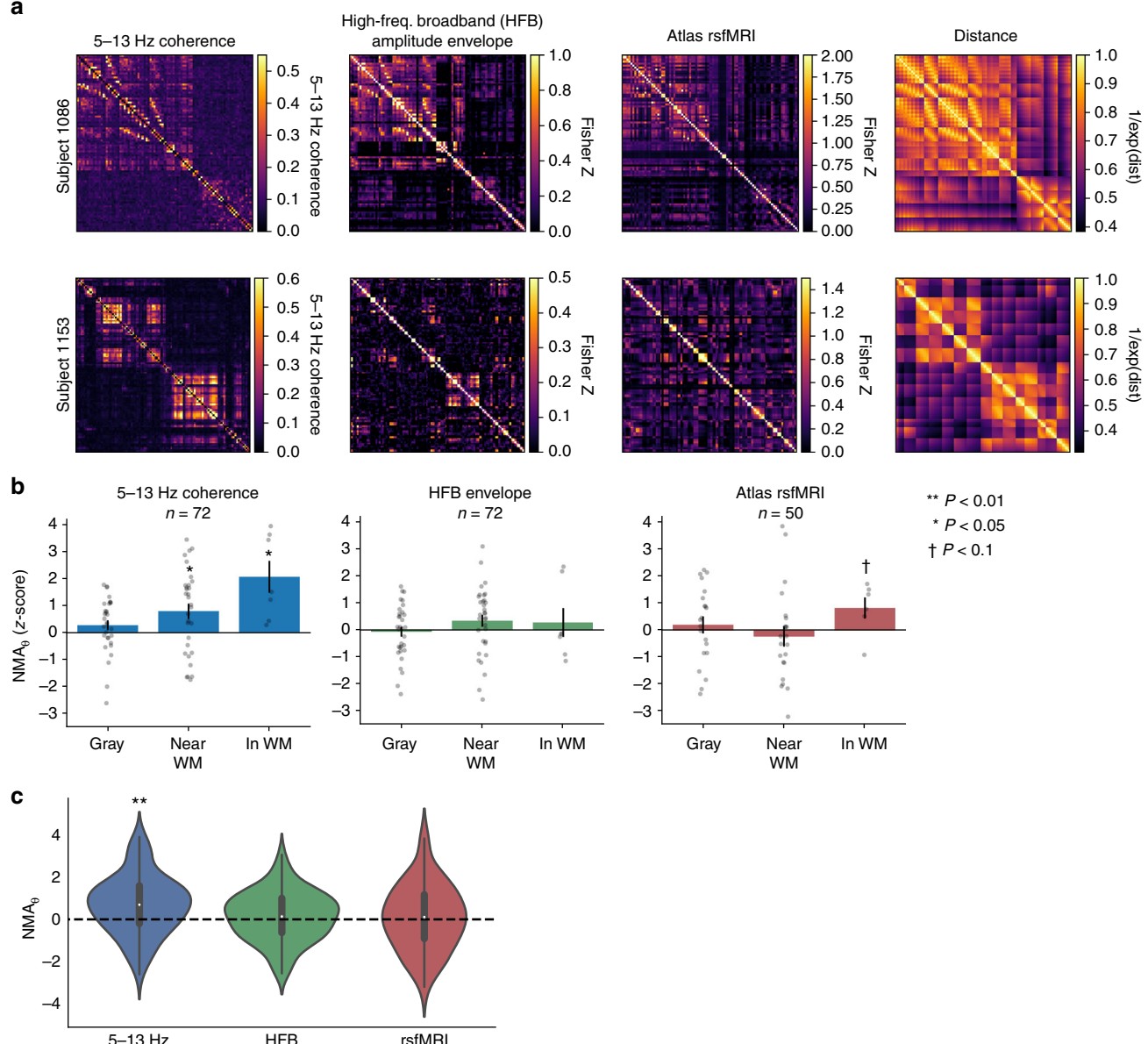

**Fig. 5** Alternative measures of connectivity. **a** Example adjacency matrices for two subjects, reflecting functional connectivity from low-frequency (5–13 Hz) coherence, correlated high-frequency broadband envelope (HFB; 50–200 Hz), and atlas-based fMRI (rsfMRI). Matrices are organized as in Fig. 2a. For reference, adjacency matrices of linearized Euclidean distance are shown at the far right. Colormap ranges are selected to visually emphasize network structure. Values in the HFB envelope and rsfMRI networks are Fisher $z$-transformed correlation coefficients. **b** $NMA_\theta$ is computed as in Figs. 2 and 3, using adjacency matrices for each of the three measures. $NMA_\theta$ was binned by distance from white matter, organized as in Fig. 3b. In addition to significant $NMA_\theta$ in and near white matter using 5–13 Hz coherence networks ($P < 0.05$), we noted marginally significant $NMA_\theta$ in white matter using atlas-based rsfMRI networks ($P < 0.1$). **c** Distribution of $NMA_\theta$ for all stimulation electrodes regardless of distance from white matter. $NMA_\theta$ is significantly greater than zero at the group level for 5–13 Hz coherence (one-sample $t$-test, $t(71) = 4.18$, $P = 8.2 \times 10^{-5}$). Note that the total count of stimulation electrodes is lower for rsfMRI connectivity ($n = 50$) analyses, due to subjects where atlas-based rsfMRI could not be extracted for a stimulation electrode; see Methods for details. Error bars show ±1 SEM; *$P < 0.05$; **$P < 0.01$

power decreases for seven stimulation sites and power increases for three stimulation sites, a total count that significantly exceeds the expected false positive rate (binomial test, FDR-corrected $P = 0.009$; Fig. 7b).

Taken together, functional connectivity measured by low-frequency coherence significantly predicts stimulation-evoked power in the theta band, but not induced power at higher frequencies. On average, HFB envelope networks do not significantly correlate with evoked changes in any frequency band, even HFB power itself. However, the dynamics of

stimulation appear to be more complex in this high-frequency band; HFB power is often decreased by stimulation—unlike the theta response—and for a significant number of stimulation sites, both low-frequency coherence and HFB functional connectivity predict where in the brain such decreases are observed.

## Discussion

We set out to test a fundamentally simple hypothesis: do functional connections in the brain predict how focal electrical

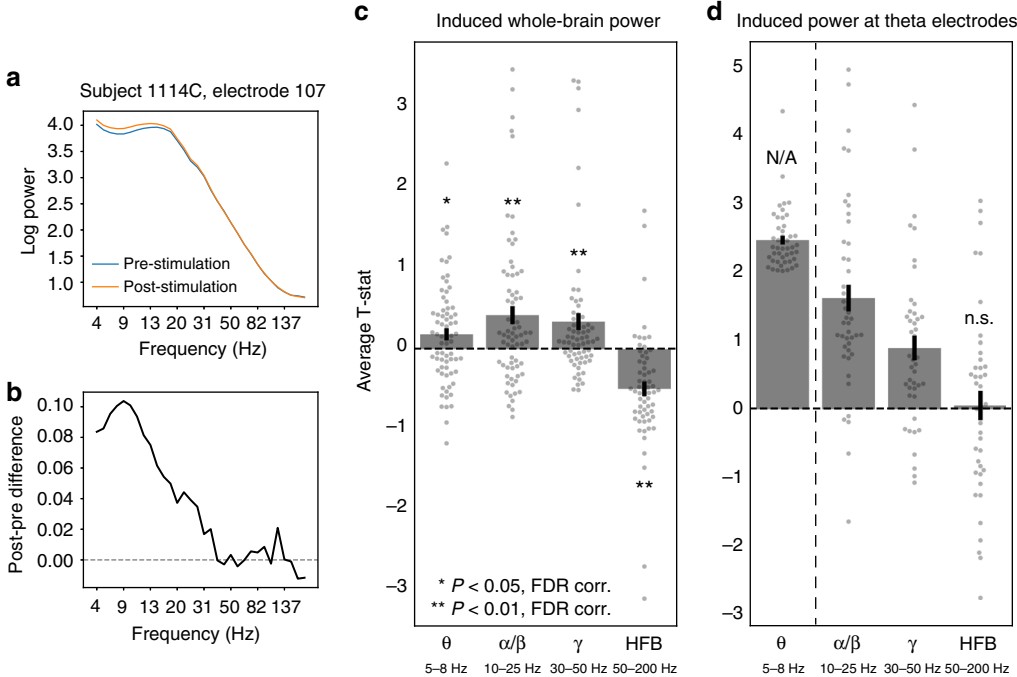

**Fig. 6** Stimulation-induced power across frequency bands. **a** Stimulation-induced power spectrogram for an example electrode from a single subject (stimulation in left MTL white matter, recording electrode in left inferior parietal cortex). **b** Post- minus pre-stimulation difference in power from the electrode in **a**. **c** Whole-brain stimulation-induced power was measured by computing a $t$-statistic on the pre- vs. post-stimulation spectral power at each electrode in a subject, and then averaging across electrodes to get an estimate of whole-brain change in power. On average, stimulation elevated whole-brain power in the theta (5–8 Hz), alpha/beta (10–25 Hz), and gamma (30–50 Hz) bands (one-sample $t$-test, FDR-corrected $P < 0.05$). Stimulation decreased power in the HFB range (50–200 Hz). **d** For each subject/stimulation site, electrodes were classified by whether they exhibited a significant ($t > 2$) change in theta power induced by stimulation (at least 1 theta-responsive electrode was found for 47 of the 72 stimulation sites). The stimulation-induced change at higher frequencies was computed for this subset of electrodes, to determine whether the power response was specific to theta. Across all subjects/stimulation sites, increased power was observed in the alpha/beta and gamma range at theta-responsive electrodes, but no effect was observed in the HFB range (one-sample $t$-test, $t(46) = 0.25$). The bar for induced theta power is delineated by a dashed line and shown as a reference only, since theta power was the basis for selecting these electrodes for further analysis. Error bars show ±1 SEM; *corr. $P < 0.05$; **corr. $P < 0.01$

stimulation flows from one region to another? Though critical to the future of brain stimulation and therapeutic development, this hypothesis has not seen rigorous testing. Prior studies indicate that connectivity plays a role in how stimulation events perturb distant brain regions[13,14,16,29], but fundamental assumptions of graph-theoretic models remain untested[17]. More broadly, no prior studies have addressed whether iEEG-based functional connectivity indicates anything about causal relationships in the brain, or whether is it merely a correlative measure. Here we specifically tested a hypothesis about the effects of stimulation on theta power, given an especially rich literature showing the cognitive relevance of theta oscillations[39–42]. To account for possible dynamics outside this range, we extended several key analyses to alpha/beta, gamma, and HFB power, and further considered whether functionally derived measures of connectivity better capture the effects of stimulation-induced power.

We discovered that (1) modulation of theta power is correlated with functional connectivity, particularly if stimulation occurred in or near white matter, (2) stronger functional connections yield greater theta power increases, (3) low-frequency coherence better predicts downstream increases in theta power than HFB envelope or rsfMRI networks, and (4) in specific cases HFB envelope networks do succeed in predicting modulations in HFB power. These results suggest that stimulation evokes a heterogenous mixture of effects across frequencies, and that functional networks may best predict the frequency on which they were based.

The meaning of functional connectivity is a subject of considerable debate. Correlated activity between two parts of the

brain may reflect direct connection between the two, an indirect connection through a third region, or the activity of a third region independently driving activity in each[43]. Though most neuroscientists are aware of such limitations, functional connectivity is often implicitly treated as a measure of causality nonetheless. Our use of targeted stimulation allowed us to test whether this implicit assumption is true. Our results generally support the idea that functional connectivity indicates causal relations in the brain; when stimulation occurs in or near white matter, we could predict where power changes would occur based on distance-independent measures of low-frequency functional connections. This finding aligns with observations that intrinsic functional connectivity in MRI is constrained by white matter anatomy[44]. However, substantial variance in power modulation remained unexplained by connectivity, and we also showed that propagation of gray matter stimulation—still rich with functional connections—cannot be predicted in the same way.

HFB amplitude envelope networks and atlas-based rsfMRI networks failed to strongly predict remote changes in theta power. However, earlier reports suggest that these functionally relevant measures do correlate with CCEPs and changes in cortical excitability[14,36]. To explain this discrepancy, we note that there are several key differences between those reports and the present study. First, we solely examined the effect of MTL stimulation, which has a distinct architecture that may affect how stimulation propagates to other regions—effects of stimulation at the cortical surface, as in prior studies, could differ markedly. Relatedly, we used stimulation amplitudes that are lower than

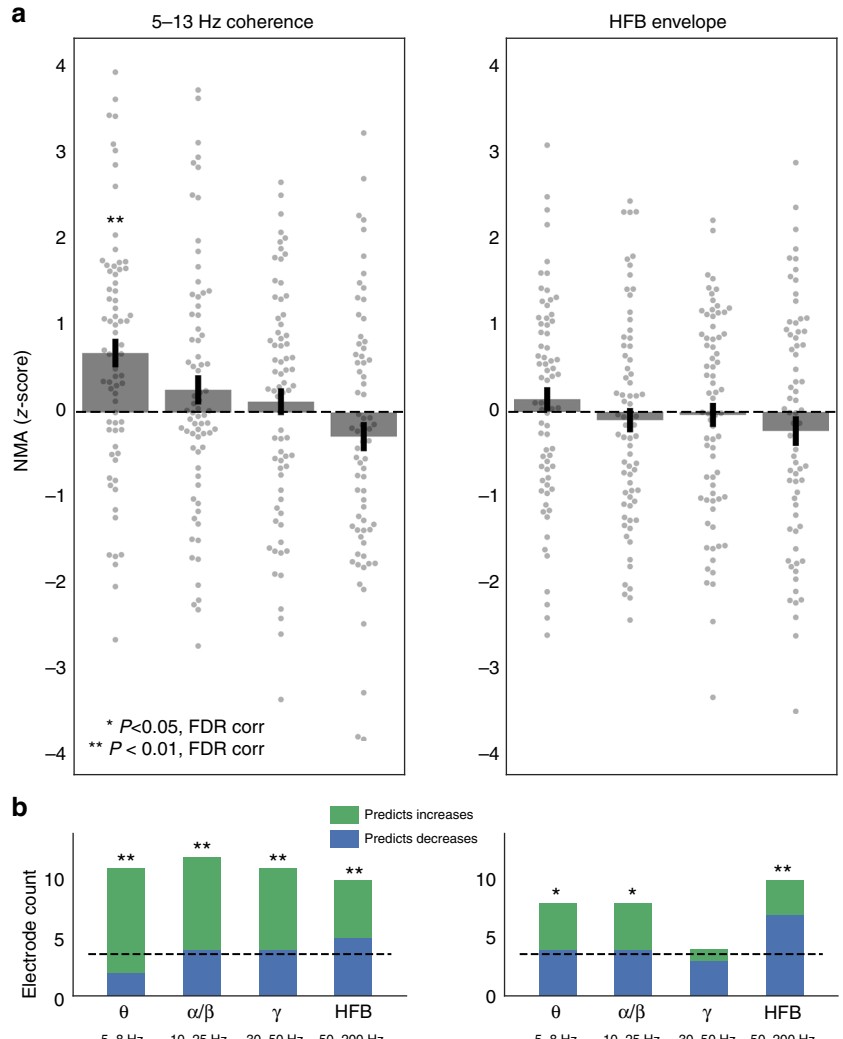

**Fig. 7** Power response at higher frequencies. **a** The average NMA—reflecting the degree to which functional connectivity predicts changes in spectral power—was computed for theta (5–8 Hz), alpha/beta (10–25 Hz), gamma (30–50 Hz), and HFB (50–200 Hz) bands for all 72 MTL stimulation sites. Functional connectivity was measured as 5–13 Hz coherence (left) and HFB amplitude correlation (right; see Methods for details). Across all stimulation sites, 5–13 Hz coherence significantly predicted changes in theta power, as demonstrated in Fig. 2 (one-sample t-test, $t(71) = 4.18$, corrected $P < 0.01$). HFB amplitude correlations did not significantly predict power changes in any band. **b** To account for the fact that connectivity could predict decreases or increases in power, each stimulation site was classified by whether its functional connectivity profile significantly predicted decreases (blue) or increases (green) power (two-tailed $P < 0.05$). The count of significant stimulation sites are depicted as stacked bars; the expected false positive rate ($P = 0.05$) is indicated as a dashed line. Note that though HFB amplitude correlations do not significantly predict changes in HFB power on average (**a**), the count of stimulation sites where connectivity significantly predicts changes in HFB power is significantly greater than chance (7 decreases, 3 increases; binomial test, corrected $P < 0.01$). Error bars show ±1 SEM; *corr. $P < 0.05$; **corr. $P < 0.01$

those typically used at the cortical surface (<2 vs. >4 mA). Finally, while HFB envelop networks did not successfully predict remote changes in theta power, they more accurately correlated with remote decreases in HFB power—it is possible that networks based on measures of cortical activation are better predictors of how stimulation affects those same measures.

In this study, we also assessed the relationship between stimulation and the network topology of a targeted region. Specifically, we asked whether the downstream effects of stimulation differed between hubs and non-hubs, reflecting regions that are richly or sparsely connected. Counterintuitively, we found that stimulation of non-hubs yielded greater increases in theta power at downstream sites. It is possible that (1) hub stimulation does result in greater distributed power changes, but outside the theta band, or (2) hub stimulation results in a dispersal or blunting

effect, causing widespread change but limiting the magnitude of the effect at any single downstream site. Such a result is plausible if there is an interaction between the underlying brain structure and the effect of stimulation—it has been demonstrated that stimulation less effectively activates large-diameter axons, for example[45]. Furthermore, principles of network control theory postulate that stimulation of sparsely connected regions can be efficacious for moving the brain to "difficult-to-reach" states, or states that require significant cognitive effort to achieve[17,18,30]. However, the mapping between spectral power and "brain states" in a cognitive sense remains unclear; further empirical and theoretical work should aim to clarify how control theoretic predictions can be tested with common intracranial techniques.

The findings from this study could be extended in several ways. A recent study by Keller et al.[36] asked whether a multivariate

model could predict how direct brain stimulation alters remote cortical excitability. A similar approach could be adopted with these data, wherein multimodal measures of connectivity—e.g. coherence, HFB envelop, and white matter proximity—could be used to predict the stimulation response across locations and frequency ranges. Such an approach could reveal relationships that were obscured by the univariate methods in this manuscript; gray matter targets, for instance, may induce widespread, connectivity-related changes in specific frequencies that are predictable by a weighted combination of functional networks. Additionally, our study as designed was agnostic to the directionality of induced effects; especially, in the setting of direct white matter stimulation, we expect that our results reflect a combination of prodromic and antidromic propagation. In other words, stimulation of MTL structures is potentially inducing activity in input and output regions, though the undirected measures of functional connectivity used here are unable to tease those effects apart.

We solely analyzed stimulation through the lens of changes in brain physiology. However, with an eye toward the eventual therapeutic use of stimulation, the results here begin to bridge prior studies of stimulation and behavior with underlying neural mechanisms. A recent study reported decreases in episodic memory performance during stimulation at certain times, associated with increases in cortical theta power[3]. Additionally, memory performance was noted to increase with theta-burst stimulation of the perforant path, a major white matter tract of the MTL[4]. Deep brain stimulation targeted to white matter tracts has also been shown to improve outcomes in treatment-resistant depression[9]. Collectively, these findings are supported by the results here—white matter stimulation appears to evoke remote increases in neural activity. Few studies have deeply examined stimulation-induced changes in physiology with behavioral enhancement, though our approach outlined here enables us to do exactly that in future work.

Here we demonstrated that functional connections in the human brain inform how stimulation evokes remote changes in neural activity. This is powerful new evidence that even in the absence of knowledge about an individual's structural connectome, functional connectivity can reflect causality in the brain —a finding with significant implications for how neuroscientists interpret inter-regional correlations of neural activity. Furthermore, by showing that stimulation-evoked changes interact with the functional hubness of a targeted site, we provided a critical data point for the application of network control theory to real-world brain dynamics.

## Methods

**Participants**. Twenty-six patients with medication-resistant epilepsy underwent a surgical procedure to implant subdural platinum recording contacts on the cortical surface and within brain parenchyma. Contacts were placed so as to best localize epileptic regions. Data reported were collected at eight hospitals over 4 years (2015–2018): Thomas Jefferson University Hospital (Philadelphia, PA); University of Texas Southwestern Medical Center (Dallas, TX); Emory University Hospital (Atlanta, GA); Dartmouth-Hitchcock Medical Center (Lebanon, NH); Hospital of the University of Pennsylvania (Philadelphia, PA); Mayo Clinic (Rochester, MN); National Institutes of Health (Bethesda, MD); and Columbia University Hospital (New York, NY). Prior to data collection, our research protocol was approved by the Institutional Review Board at participating hospitals, and informed consent was obtained from each participant.

**Electrocorticographic recordings**. iEEG signal was recorded using depth electrodes (contacts spaced 3.5–10 mm apart) using recording systems at each clinical site. iEEG systems included DeltaMed XlTek (Natus), Grass Telefactor, and Nihon-Kohden EEG systems. Signals were sampled at 500, 1000, or 1600 Hz, depending on hardware restrictions and considerations of clinical application. Signals recorded at individual electrodes were first referenced to a common contact placed intracranially, on the scalp, or mastoid process. To eliminate potentially confounding large-scale artifacts and noise on the reference channel, we next re-referenced the data using a bipolar montage. Channels exhibiting highly non-physiologic signal due to damage or misplacement were excluded prior to re-referencing. The resulting bipolar time series was treated as a virtual electrode and used in all subsequent analysis. Raw electrophysiogical data and analysis code used in this study are freely available at http://memory.psych.upenn.edu/Electrophysiological_Data.

**Anatomical localization**. To precisely localize MTL depth electrodes, hippocampal subfields and MTL cortices were automatically labeled in a pre-implant, T2-weighted MRI using the automatic segmentation of hippocampal subfields multi-atlas segmentation method[46]. Post-implant computed tomography (CT) images were coregistered with presurgical T1- and T2-weighted structural scans with Advanced Normalization Tools[47]. MTL depth electrodes that were visible on CT scans were then localized within MTL subregions (including white matter) by neuroradiologists with expertise in MTL anatomy. All localizations in this manuscript refer to the bipolar midpoint of two recording contacts or the anode/cathode stimulation contacts.

**Functional connectivity estimation**. To obtain coherence values between electrode pairs, we used the MNE Python software package[48], a collection of tools and processing pipelines for analyzing EEG data. The coherence ($C_{xy}$) between two signals is the normalized cross-spectral density (Eq. 1); this can be thought of as the consistency of phase differences between signals at two electrodes, weighted by the correlated change in spectral power at both sites.

$$C_{xy} = \left| \frac{S_{xy}}{S_{xx}S_{yy}} \right| \tag{1}$$

Where $S_{xy}$ is the cross-spectral density between signals at electrodes $x$ and $y$; $S_{xx}$ and $S_{yy}$ are the auto-spectral densities at each electrode. Consistent with other studies of EEG coherence[49,50], we used the multitaper method to estimate spectral density. We used a time-bandwidth product of 4 and a maximum of 8 tapers (tapers with spectral energy < 0.9 were removed), computing coherence for frequencies between 4 and 50 Hz, avoiding the 60 Hz frequency range that may be contaminated by line noise. Inter-electrode coherences were computed for a series of 1-s windows extracted sequentially from 10-s baseline periods of a non-stimulation task, in which subjects wait passively before beginning a verbal free-recall task. Each subject typically had 24–72 such baseline periods, but all had a minimum of 10 (i.e. the minimum total number of windows used for network estimation was 100). To construct the low-frequency networks used in the majority of this paper, cross-spectra were first averaged across all baseline period windows, normalized by the average power spectra, and then averaged between 5 and 13 Hz. For the analysis in Fig. 4e, networks are constructed for each frequency between 4 and 50 Hz with no averaging over bands.

**Stimulation paradigm**. At the start of each session, we determined the safe amplitude for stimulation using a mapping procedure in which stimulation was applied at 0.5 mA, while a neurologist monitored for afterdischarges. This procedure was repeated, incrementing the amplitude in steps of 0.5 mA, up to a maximum of 1.5 mA (chosen to be below the afterdischarge threshold and below accepted safety limits for charge density[51]). For each stimulation session, we passed electrical current through a single pair of adjacent electrode contacts in the MTL. Stimulation was delivered using charge-balanced biphasic rectangular pulses (pulse width = 300 μs) at (10, 25, 50, 100, or 200) Hz frequency and (0.25–2.00) mA amplitude (0.25 mA steps) for 500 ms, with a minimum of 3 s between stimulation events. During a session, subjects were instructed to sit quietly and did not perform any task. An average of 2.7 stimulation sites were selected for each subject, with a minimum of 240 trials delivered for each. In a typical stimulation session, a given target would receive 360 total stimulation events, in blocks of 60 trials at each amplitude, with 12 randomly spaced trials at each frequency within the block (Fig. 1d). For all analyses in the main text, effects were aggregated across stimulation parameters; see Supplementary Figure 2 for consideration of stimulation frequency and amplitude.

In most subjects, a post-stimulation voltage deflection artifact briefly contaminates a subset of recording contacts. To identify and remove channels exhibiting this artifact, the average voltage in the 350 ms prior to stimulation is compared with a paired t-test to the average voltage in the 350 ms after stimulation, across all trials, for each channel. The same procedure is done with a levene test for different variances. Any electrode with a significantly different pre- vs. post-mean voltage or voltage variance (P < 0.01) is excluded from further analysis (see "Estimating NMA"). On average, this procedure excludes 28% of channels. Regardless of stimulation artifact, any bipolar pair is excluded from analysis if it shares a common contact with the stimulated pair. See Supplementary Figure 4 for a representative example of this artifact.

**Spectral power analysis**. We used the multitaper method to assess spectral power in the pre- and post-stimulation intervals (−950 to −50 ms relative to stimulation onset, and +50 to +950 ms after stimulation offset; Fig. 1b). We avoided the Morlet wavelet method to obviate the need for buffer periods that extend into the

stimulation window. As in "Functional connectivity estimation," we used the MNE Python software package. For each trial, theta power was taken as the average PSD from 5–8 Hz, using a time-bandwith product of 4 and excluding tapers with <90% spectral concentration. To compute a $t$-statistic at each electrode, the pre- vs. post-log-transformed power values were compared with a paired $t$-test (Figs. 1g and 2b). We avoid calculating significances for individual electrodes because sequential trials are non-independent events; $t$-statistics are only used for later correlation analysis (see "Estimating NMA").

For analyses that considered spectral power at higher frequencies (Figs. 6 and 7), we used the following bands: alpha/beta (10–25 Hz), gamma (30–50 Hz), and HFB (50–200 Hz). Power was otherwise computed exactly as described for theta. To measure whole-brain evoked power (Fig. 6), we took the average $t$-statistic across all electrodes in each subject's brain, subject to the same exclusion criteria described in "Estimating NMA." Additionally, we excluded electrodes with $t$-statistics > 10 from the whole-brain average, to account for raw power values that are potentially corrupted by post-stimulation artifact, which survives our exclusion procedure (their inclusion does not notably change the main results).

**Estimating NMA**. To examine the relationship between stimulation and functional connectivity, we developed an index that reflects the correlation between theta power modulation and connectivity, independent of distance. To do this, we first construct low-frequency (5–13 Hz) networks as described in "Functional connectivity estimation," and take the logit transform to linearize coherence values that fall between 0 and 1. We also construct adjacency matrices that reflect the normalized Euclidean distance between all possible pairs of electrodes (Fig. 2a), and linearize the distances by taking the reciprocal of their exponential (i.e. a Euclidean distance of zero would correspond to 1.0). For each stimulated electrode, we take that electrode's distance and connectivity to all other electrodes as predictors of the theta power $t$-statistic (see "Spectral power analysis") in a multiple linear regression. This controls for the effect of distance from a stimulation target, which is correlated with power and functional connectivity. Next, we permute the order of the predictors 1000 times and re-run the regression for each. The true coefficient for functional connectivity is compared to the distribution of null coefficients to obtain a $z$-score and $P$-value for each stimulation site. The $z$-score is referred to as the NMA.

Prior to computing $NMA_\theta$, we excluded electrodes placed in the seizure onset zone or exhibiting significant inter-ictal spiking, as determined by a clinician. Electrodes with high post-stimulation artifact (see "Stimulation paradigm"), and stimulated electrodes themselves, were also excluded. Subjects were discarded if <10 electrodes remained after all exclusions.

To analyze the relationship between $NMA_\theta$ and white matter category (Fig. 3), we first binned electrodes according to their distance from nearest white matter. Distance were measured as the linearized Euclidean distance from a stimulation electrode (i.e. bipolar midpoint of the anode/cathode) to the nearest vertex of that subject's Freesurfer white matter segmentation[52] based on T1 MRI. The 50th percentile of white matter distances marked the division between stimulation electrodes categorized as "near" white matter vs. in gray matter. Seven stimulation electrodes were identified by expert neuroradiologists as being placed within white matter (see Supplementary Figure 1 for exact placements). To ask whether $NMA_\theta$ increases with white matter category, we permuted the white matter labels for each electrode 1000 times and took the minimum $t$-statistic between gray vs. near and near vs. in-white categories at each permutation. We then compared the minimum $t$-statistic in the true data to the distribution of null statistics to generate a $P$-value.

**Network properties of stimulation**. To determine how the network structure of a stimulation site affected downstream alterations in theta power (Fig. 4), we first analyzed the relationship between pre- vs. post-stimulation theta power and the strength of functional connectivity to a stimulation site (Fig. 4a, b). For each stimulation site with a significant $NMA_\theta$ ($P < 0.05$), we ranked all other electrodes by the strength of their functional connectivity to that site, residualized on Euclidean distance ($e^{-dist}$). We then took the average power $t$-statistic (see "Spectral power analysis") across the five strongest-connected sites and the five weakest-connected sites, to assess whether theta power changes correlated with the strength of a functional connection.

To assess whether the effects of stimulation differ between hubs and non-hubs (Fig. 4c, d), we measured the node strength[53] for each stimulation site in or near white matter ($n = 40$), using our low-frequency coherence networks (see "Functional connectivity estimation"). The node strength is the summed sum of all connection strengths to a given node (for this paper, we normalized node strength by the total number of possible connections for a given site, yielding strengths in the range from 0 to 1). For all stimulation sites, we binned hub scores by tercile, and took the highest tercile as "strong hubs," the weakest tercile as "weak hubs" ($n = 13$ for each). For stimulation at all strong and weak hubs, we took the average power $t$-statistic for the five strongest-connected electrodes. These values were used to assess whether hub stimulation tends to cause greater power changes in connected regions. The relationship between coherence frequency and NMA (Fig. 4e) was assessed by re-estimating the $NMA_\theta$ (see "Estimating network-mediated activation") using spectral coherence networks observed for each frequency between 4 and 50 Hz, spaced by 1 Hz, for all stimulation electrodes placed within or near white matter. The average $NMA_\theta$ across sites/subjects was

one-sample $t$-tested against zero and $P$-values were FDR-corrected for multiple comparisons (corrected $P < 0.05$). For visualization purposes only, the displayed $NMA_\theta$/frequency curve was smoothed with a three-point moving average window.

**Alternative connectivity metrics**. HFB amplitude envelop correlation: Networks of correlated HFB (50–200 Hz in this manuscript) amplitude envelops were computed in a manner similar to Foster et al.[23]. The general approach is to low-pass filter HFB spectral power during a resting period, and the resulting time series are correlated between recording electrodes to construct an adjacency matrix. Specifically, in a non-stimulation memory task, we extracted 240-s (4 min) resting periods between any task events. Resting periods were identified by searching for the maximum amount of time between task events; in some subjects, 240-s intervals were not available but time series were still extracted for that length in the best-possible period. Signals were bipolar re-referenced and notch filtered, sequentially band-passed in 10 Hz windows from 50 to 200 Hz, Hilbert transformed and normalized to the mean amplitude, and then averaged across bands. Finally, to estimate the slow variation in this signal as a basis for inter-regional correlation, we low-pass filtered the HFB amplitude (<1 Hz), and computed the Pearson correlation coefficient between the resulting signals between all possible pairs of electrodes within a subject, yielding an adjacency matrix of correlations. The resulting correlations were Fisher $z$-transformed and then used as predictors of modulations in power (see "Estimating NMA").

Atlas-based rsfMRI: We used an independent dataset of rsfMRI from the Human Connectome Project (HCP)[54] to estimate functional connectivity between recording sites in each patient. For each patient, we mapped the location of subdural and depth electrodes to the HCP grayordinate space[55]. For subdural electrodes, we assigned vertices on the cortical surface mesh within 3 mm (geodesic distance) of each recording site to a region of interest (ROI). The coordinates in the native space of each subject were then mapped to the standard fs_LR mesh (i.e., HCP surface space). The location of subcortical contacts in native space were transformed to MNI space using Advanced Normalizations Tools[56], with spherical ROIs centered at each bipolar midpoint. Adjacency matrices for each subject were constructed by computing the average connectivity (Fisher-transformed time-series correlations) between all grayordinates from each pairwise combination of ROIs, provided by the group-averaged ($n = 897$ subjects) dense connectome. These adjacency matrices were subsequently used to determine whether fMRI defined networks provide a scaffold for the propagation of brain-wide theta power following DES.

## Data availability

Raw electrophysiogical data and analysis code used in this study are freely available at http://memory.psych.upenn.edu/Electrophysiological_Data

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

## Acknowledgements

We thank Blackrock Microsystems for providing neural recording equipment. This work was supported by the DARPA Restoring Active Memory (RAM) program (Cooperative Agreement N66001-14-2-4032), as well as National Institutes of Health grant MH55687 and T32NS091006. We are indebted to all patients who have selflessly volunteered their time to participate in our study. The views, opinions, and/or findings contained in this material are those of the authors and should not be interpreted as representing the official views or policies of the Department of Defense or the U.S. Government. We also thank Drs. Youssef Ezzyat, Christoph Weidemann, Nora Herweg, Danielle Bassett, and Geoffrey Aguirre for providing valuable feedback on this work. Data were provided in part by the Human Connectome Project, WU-Minn Consortium (Principal Investigators: David Van Essen and Kamil Ugurbil; 1U54MH091657) funded by the 16 NIH Institutes and Centers that support the NIH Blueprint for Neuroscience Research; and by the McDonnell Center for Systems Neuroscience at Washington University.

## Author contributions

E.A.S., M.J.K., and D.S.R. designed the study; E.A.S. conceived, planned, and executed all data analyses, J.E.K. analyzed data, and E.S. wrote the paper. J.M.S., R. Gorniak, and S.D. performed anatomical localization of depth electrodes. M.R.S., G.W., B.L., R. Gross, B.C.J., C.S.I., K.A.Z., S.S., and S.A.S. recruited subjects, collected data, and performed clinical duties associated with data collection including neurosurgical procedures or patient monitoring.

## Additional information

**Competing interests:** M.J.K. and D.S.R. have started a company, Nia Therapeutics, LLC ("Nia"), intended to develop and commercialize brain stimulation therapies for memory restoration. Each of them holds >5% equity interest in Nia. R. Gross serves as a consultant to Medtronic, which is a subcontractor on the RAM project. R. Gross receives compensation for these services. The terms of this arrangement have been reviewed and approved by Emory University in accordance with its conflict of interest policies. The remaining authors declare no competing interests.

