## [Peer Review File · Nature Communications]

Reviewers' comments:

Reviewer #1 (Remarks to the Author):

In this paper, Solomon and colleagues studied 26 patients implanted with intracranial electrodes across their brains. They stimulated few MTL sites in each subject's brain with frequencies 10, 25, 50, 100, or 200 Hz frequency and varying intensity (0.25 to 2.00mA) while recording from all other electrodes. They found that low-frequency (5-13 Hz) spectral coherence between electrodes predicted stimulation-evoked changes in theta (5-8 Hz) power in those electrodes. In other words, if a pair of brain regions have coherent low frequency oscillations at baseline, stimulating one will change the power of slow frequency oscillations in the other. Moreover, this effect will be better observed if the stimulation target is within or near the white matter.

The work is unique and important - as many other excellent papers published from Kahana's lab. It provides important and novel information about the propagation of injected electrical discharges in the human brain. The research is also important to the field of resting/intrinsic connectivity as it demonstrates how inter-regional coherent spontaneous activity have predictive utility for knowing where the neuromodulatory effects of electrical perturbation will be seen.

I hope my suggestions will make this paper even stronger for the large readership of Nature Communication:

- It will be helpful to bring Fig 2E to Fig 1. Suppl Table 2 suggests that the majority of these sites were within the hippocampus. Current Fig 1A shows a figurative site of stimulation in the left middle temporal gyrus (a grid is shown) whereas to my understanding most of the stimulations were through depth electrodes in the hippocampus proper. This could be clarified.
- It should be helpful to make it clearer with tables or figures how many stimulations were performed within each of the frequencies in 10-200 Hz range and whether there was any frequency effect.
- Functional connectivity was defined as 5-13 Hz coherence, and authors state (Line 172) that "functional networks constructed from high frequencies (> 13 Hz) are not predictive of stimulation-induced theta activity". While I agree that "functional connectivity" is a loose term, and any random measure could be chosen to denote such "connectivity", I find the 5-13Hz coherence severely limiting. The same applies to gamma-gamma coherence. If we are to define functional connectivity as the intrinsic connectivity of two regions that are engaged together in a cognitive conditions, then we must show that the two "connected" regions have similar functionalities. Using this stringent criterion, and akin to fMRI BOLD functional connectivity maps in the same subjects, we have previously shown that the correlated infra-slow fluctuations of the high frequency broadband (high gamma, 70-150Hz) should be taken as the metric of such connectivity - I urge the authors to look at Foster et al (Neuron 2015 – see Fig 5E) and Kucyi et al (J Neurosci 2018) and summarized in Fox et (TICS 2018 – sited already in this manuscript). Coherence in 5-13Hz is a weak measure and I am not sure if it really reflects "functional" connectivity. I am familiar with the authors previous publications on coherent 5-13Hz (Nat Comm 2017), and in none of those publications the functional connectivity in each subject was verified with anatomical and physiological selectivity of the responses during a cognitive task + fMRI resting state in et same subject. I do not think the authors should redo all their analyses, but they need to provide analyziz of <1Hz envelope of HFB instead of gamma-gamma coherence measures they have performed. This is especially important because authors identified connectivity on the basis of coherence in slow frequency and then showed that the slow frequency power only changes with stimulation in those areas that are "connected". This seems like double dipping and circular argument even though the two samples are from entirely separate sessions (rest and recall).
- While the authors have performed several control analyses to improve confidence that the effect

was strongest for white matter electrodes, there remain at least two issues: Was one frequency of stimulation used at each electrode site (the protocol is not clear in the Methods)? If so, were the stimulation frequencies used at white matter sites comparable to those used at other sites? The authors state that "Inter-electrode coherences were computed for a series of 1-second windows (minimum of 10 windows per subject) extracted from the baseline period of a non-stimulation task, in which subjects wait passively before beginning a verbal free-recall task." Does this mean that some baseline periods were only 10 seconds long whereas others were much longer? Was there a difference between patients who had white matter electrodes versus those who did not?

- Could the authors provide more information/analysis of the anatomical locations of target electrodes? Did they include all white matter electrodes? Which regions comprised the "strong connections" (Fig 4B)? I understand that this differs between patients, but a summary would be helpful.

- Did the authors monitor for changes in behavior or subjective experience during stimulation? Please mention it briefly in the manuscript

- In the Discussion, the authors state that "no prior studies have addressed whether iEEG-based functional connectivity indicates anything about causal relationships in the brain, or whether it is merely a correlative measure." This is disappointing. They should acknowledge previous work showing that fMRI functional connectivity is predictive of the effects of CCEPs (specially Keller et al 2011 PNAS; Shine et al 2017 J Neurosci).

- Betzel, R. F. et al. reference details missing.

- Line 84-87: These networks reflect correlated low-frequency activity between all possible pairs of electrodes in a subject. What does this mean?

- Precise number of stimulation targets: Why did we end up with 16 sites in the final analysis? How many subjects are these from? Can the authors be more clear (and transparent) about the selection of their stimulation target areas for the analysis. 40 sites were inside or near the white matter: But then the authors state (line 134) that only 16 stimulation sites exhibited significant TMI.

- Paper refers to 33 electrodes near and 7 inside white matter and state that TMI for sites near or in white matter was significant. How much of this effect is due to the 7? What if you take out the 7? I personally think white matter targets should be analyzed completely separately. Stimulating inside the white matter in the proximity of MTL could be targeting connections coming from remote sites and stimulation could be inducing anti-dromic effects. Authors should please comment on antidromic effect versus prodromic. General reader would otherwise believe that the effects of stimulation always follow the direction of the axons. It could very well go antidromically as previously shown in the literature.

- Line 122: It is very puzzling that theta activity induced by gray matter stimulation is uncorrelated with functional connectivity to remote sites. But then the conclusions in the paper are stated as if the gray matter stimulations really mattered –though in lesser degrees. This needs to be discussed thoroughly.

- Line 130: Please remove the mention of DIRECTIONALITY. functional connectivity does not necessarily denote directionality. Similarly, line 201 states that "results generally support the idea that functional connectivity indicates causality". This is very controversial and depends on how one defines causality. Please explain this and/or take away the notion of causality. It is really not that relevant to the findings of the study anyway.

- Line 147: Regions with strong connectivity to the rest of the brain, are generally less capable of modulating the brain's overall state versus non-hubs. This seems not right. How was the "hob-ness" measured? Can the authors please explain?

- Line 220 says: the findings "bridge prior studies of stimulation and behavior with underlying neural mechanisms". I think the authors should water down this statement. I do not think the current study reveal a mechanistic explanation for how stimulation and behavior are related.

Overall, I congratulate the team for producing this excellent piece and I hope that my comments could improve its quality even further.

Reviewer #2 (Remarks to the Author):

The authors present a well written manuscript describing low frequency (theta and alpha) activity in the mesial temporal lobe and how the propagation of these events relate to functional connectivity methods from a series of 26 neurosurgical patients. While the topic is timely, the analyses appear well motivated and the manuscript is well written, a number of concerns dampen my enthusiasm for publication in Nature Communications as outlined below:

1. My main concern regards the relationship between electrical stimulation and induction of the theta rhythm. While Figure 1 shows an example of pre stim and post-stim theta, it is not clear whether this represents a broad band increase in power (as would occur in the case of a stimulation-induced afterdischarge) or the authors hypothesis that this phenomenon is selective to lower frequencies. Without a spectrogram depiction both for the individual case as well as some summary of the entire data set (as attempted in figure 1E), it is not possible to determine the relationship between specific frequency bands and any effects demonstrated.

2. They used a range of electrical stimulation from 10-200 Hz, but do not detail what specific stimulation frequency results in the increase in theta power. If the premise here is that electrical stimulation induces theta in the first place, then it would be quite valuable to know exactly how to increase theta power in the hippocampus so results may be replicated and the field could advance. A particular concern relates to the fact that because of this, the analyses may be subject to cherry-picking the most consistent results from one particular stimulation frequency (out of the 5 different ones used). This concern further makes figure 1E difficult to interpret.

Based upon these two major concerns, the main findings here are that signals (presumably causal to stimulation) propagate to areas of connectivity. While this is demonstrated here, the findings are largely predictable. The efficacy of white matter stimulation and the differential effects of hub/non-hub and strong/weakly connected areas is intriguing, but would benefit from more detailed analyses considering correlation of DTI and alternate resting connectivity measures.

Reviewer #3 (Remarks to the Author):

In this study, Solomon et al apply electrical stimulation to the medial temporal lobe in 26 neurosurgical patients to answer the question of how external perturbation modulates brain networks. They find networks of theta coherence predicted stimulation-evoked changes in theta power, but only when stimulated electrodes were in white matter. Furthermore, they find that when network hubs were stimulated, it evoked less theta changes than when non-hubs were stimulated. This question is important to better understand the mechanism underlying brain stimulation treatments, and its potential to optimize and personalize these treatments. The study

is potentially highly impactful, but in my opinion a major revision is necessary before publication. My questions / comments are below:

1] I found the methods and amount of terms in the study to be confusing, and takes the reader away from the important findings in here. I would recommend a careful review to see if any abbreviations can be removed or simplified. TMI was difficult to understand. Additionally, for example, in figure 2, terms like BP, LFA, Logit are not clear from the figure or the legend.

2] I would recommend adding a discussion about a recent publication by Keller et al., JNeuro, 2018 as it is relevant to this work. A few sentences in the discussion would be helpful to compare this current study with the recent publication, and how it either complements or contrasts the findings and methods on how to explore/residualize distance and the relationship between functional connectivity and stimulation-induced changes observed.

3] In general, I had a very hard time determining what stimulation frequencies were used. In the methods, it states that a variety of amplitude, frequency, and duration was used, in an event-related design. This should be very clear to the reader which stimulation parameters were used for which analyses, and be clearly reflected in the figures and text. Maybe I missed it, but I was under the understanding that only theta frequency stimulation was examined, which would make sense that theta stimulation elicits theta-specific changes post-stimulation. More interesting would be to show that for other stimulation frequencies, either only theta modulation was observed, or the modulation observed was dependent on the stimulation frequency. This is an important question that should be addressed in this study and not a future study, especially because the data has been collected and strong statements are made about theta specifically, but may instead be purely dependent on the stimulation frequency applied. My apologies if this was clear but I just did not fully understand it.

4] In many ECoG studies, (like this one) often electrodes across subjects need to be grouped together in order to achieve sufficient power for an analysis. I would be curious how many subjects the 7 'within WM' electrodes came from? It would be relevant if it is from 7 patients or from 1.

5] Regarding the hub analysis, in the abstract you state, 'these power changes aligned with control-theoretic predictions of how exogenous stimulation flows through complex networks' and later 'while counterintuitive, this result is in line with the prediction of network control theory.' I think this finding warrants further discussion. Finding that stimulation of a hub region elicits less modulation than a non-hub region is the most interesting finding in this paper, in my opinion. Most people would have thought a strong hub region would have more ability to modulate regions within the network. If true, this has implications for the fields of non-invasive and invasive brain stimulation. However, the authors have minimal citations regarding this counterintuitive finding being in line with network control theory, and instead state it as if it is a given, which my guess is most readers would think otherwise. A longer discussion on this point is warranted. If space is an issue, I found paragraph 3 in the discussion to be slightly redundant.

6] Pre-stimulation theta calculation. If my understanding is correct, the authors applied repetitive stimulation trains in an event related design every 3 seconds, and used the 900ms before the stimulation train for the baseline period. Although likely enough time, it would be worthwhile to show that the train before did not influence the next train's pre-stimulation baseline period. Showing a control analysis of pre/post compared to pre/post2s (1 or 2s after the train) would be helpful and clarify this.

7] Figure 2B - different colors for SOZ and artifact so there is a clearer contrast between regions. Same for D - black lines would be clearer, as you're already using red in the figure to denote the T-stat strength.

8] Stimulation artifact. The authors state there is an artifact lasting for up to 350ms after the last

stimulation pulse. However, I wonder if this is neural vs non-neural. Can the authors show what this artifact looks like? 28% of channels were removed because of this procedure, and if neural in origin they may be removing nearly a third of the most strongly connected regions.

9] Regression procedure. I understand and applaud the authors for regressing out distance as it is clearly an important and often overlooked factor. Some analysis or citations of how distance is often correlated with strength of response would be helpful. On that note, it may not be necessary, but should at least be commented on why a multivariate regression model wasn't used to predict regions of modulation, using features including distance, coherence, and theta power, similar to the recent JNeuro paper by Keller et al. This way one can see the amt of theta power explained by anatomical distance vs coherence. Other related questions include compared to absolute distance measure, how well does functional connectivity predict TMI? (Provide the R^2 for both). And is distance colinear with functional connectivity? If so, then the resulting linear regression will be unstable and this should be addressed.

10] In general, I found some of the discussion text to be unnecessarily strong, especially around the hub finding. A softer discussion and one that offers other alternative explanations as to why the counterintuitive finding was observed, would be interesting and helpful for future studies.

11] I found the lack of animal literature in the paper to be striking. Is there really minimal studies examining the relationship between functional connectivity and causal connectivity? If so, it should be more clearly stated. If not, citations should be added.

12] The paper would benefit from a discussion about the type of stimulation applied (examining short-term plasticity), and how it contrasts with single pulse CCEP studies (probing and very temporary changes) to single train application to longer-term changes as in Keller et al paper and as is applied in TMS treatment for depression.

13] Is there a possibility the theta power increase simply reflects the evoked potential of the last pulse? It may be worth exploring what happens when you remove the first 500+ms of the last pulse in the train, as the evoked potential can certainly last up to 1s, and the slow wave can be in the range of theta.

14] What does the spectrum of power look like after the train? Is there a selective increase in theta power or do all other frequency band also see increased power? i.e. how selective is this theta power increase?

Summary of Major Revisions

Before enumerating each reviewer's comments line-by-line on the next page, we first include a high-level summary of the major revisions we made to the manuscript based on the reviewers' feedback:

1. Consideration of alternative connectivity metrics. All three reviewers asked that we consider different connectivity metrics beyond the spectral coherence measure used in the original manuscript. We wholeheartedly agree that this extension would enhance our study and broaden its appeal. Accordingly, we have added an extensive new analysis that considers two validated, functionally-relevant measures of inter-regional connectivity in the brain: (1) the correlated amplitude envelope of high-frequency broadband (HFB; specifically requested by Reviewer 1), and (2) atlas-based resting state fMRI connectivity. In a new figure (Figure 5, reproduced here on pg. 5), we demonstrate that these networks do not predict the location of evoked theta power as well as the original low-frequency coherence metric. However, we hypothesized that such networks may better correlate with evoked power at higher frequencies. In Figures 6-7 (pgs. 6 and 13 in this document), we show how stimulation appears to evoke a mixture of high-frequency power increases and decreases, but that HFB-based networks are capable of predicting the location of such changes in some subjects. While more work remains to be done, our revised manuscript suggests that the differential effects of stimulation across frequency bands can be captured by inter-regional networks constructed at different timescales.

2. Consideration of evoked power at higher frequencies. All three reviewers noted that our original manuscript focused solely on theta (5-8 Hz), ignoring potential stimulation-evoked effects at higher frequencies. While our focus on theta was deliberate – driven by a vast literature on these low-frequency oscillations – we agree that any reader would naturally ask about power changes at higher frequencies. Therefore, we have retained a focus on theta, but added a substantial new analysis considering evoked power changes in the alpha/beta, gamma, and HFB ranges (reported in Figures 6-7, pgs. 6 and 13). We found that stimulation generally tends to cause widespread increases in theta, alpha/beta, and gamma power, and decreases in HFB power. However, for specific electrodes that exhibit strong theta increases, the effect is reasonably specific to relatively low frequencies. Additionally, as discussed in point (1), low-frequency coherence tends to capture changes in theta power but not at higher frequencies. HFB-based networks do not correlate with changes in theta power, but do capture some of the heterogenous effects observed in HFB power itself.

3. Consideration of stimulation parameters. All three reviewers asked whether theta effects were specific to a certain stimulation pulse frequency. In the original manuscript, all reported effects were aggregated across stimulation amplitudes and frequencies, reflecting a neural response to stimulation pulse trains in general. In the revised manuscript, we have made changes throughout the text and added a new subpanel in Figure 1 to make our experimental design clearer. Additionally, we have added a new analysis to ask whether there are differential theta responses as a function of stimulation amplitude and frequency (Supplemental Figure 2, reproduced here on pg. 3). In summary, we found that higher-amplitude stimulation tends to evoke greater increases in theta power, but stimulation frequency had no effect on evoked theta power or the theta modulation index (TMI; renamed NMA in the revised text but within this response document we retained use of TMI for consistency with referee comments). We therefore have continued to aggregate effects across stimulation parameters, but have included this new analysis in the revised manuscript.

4. Additional discussion points. We have substantially revised the Discussion to include clearer and deeper discussion on several key points, including (1) implications of the hub stimulation response in the context of network control theory, (2) relation of the current work to recent studies on human intracranial stimulation and connectivity, and (3) integration of the new results regarding alternative connectivity metrics.

Reviewers' comments:

Reviewer #1 (Remarks to the Author):

In this paper, Solomon and colleagues studied 26 patients implanted with intracranial electrodes across their brains. They stimulated few MTL sites in each subject's brain with frequencies 10, 25, 50, 100, or 200 Hz frequency and varying intensity (0.25 to 2.00mA) while recording from all other electrodes. They found that low-frequency (5-13 Hz) spectral coherence between electrodes predicted stimulation-evoked changes in theta (5-8 Hz) power in those electrodes. In other words, if a pair of brain regions have coherent low frequency oscillations at baseline, stimulating one will change the power of slow frequency oscillations in the other. Moreover, this effect will be better observed if the stimulation target is within or near the white matter.

The work is unique and important - as many other excellent papers published from Kahana's lab. It provides important and novel information about the propagation of injected electrical discharges in the human brain. The research is also important to the field of resting/intrinsic connectivity as it demonstrates how inter-regional coherent spontaneous activity have predictive utility for knowing where the neuromodulatory effects of electrical perturbation will be seen.

I hope my suggestions will make this paper even stronger for the large readership of Nature Communication:

We truly appreciate the thoughtful, detailed, and extensive comments from the Reviewer. We have substantially incorporated this feedback into the revised manuscript and believe it has helped us craft a far better contribution to the literature. Please see below for our responses to each critique.

- It will be helpful to bring Fig 2E to Fig 1. Suppl Table 2 suggests that the majority of these sites were within the hippocampus. Current Fig 1A shows a figurative site of stimulation in the left middle temporal gyrus (a grid is shown) whereas to my understanding most of the stimulations were through depth electrodes in the hippocampus proper. This could be clarified.

Figure 2E is now displayed as a subpanel in Figure 1. The Reviewer's understanding is correct – all stimulations were through depth electrodes placed in the MTL (though they may be outside the hippocampus proper). The intent with the original schematic Figure 1A was for the arrow to be pointing through the “glass brain” to an MTL stimulation site, though the 2D projection does make this potentially confusing; we hope that incorporating Figure 2E into the revised Figure 1 helps clarify this.

- It should be helpful to make it clearer with tables or figures how many stimulations were performed within each of the frequencies in 10-200 Hz range and whether there was any frequency effect.

A new subpanel has been added to Figure 1 which describes the exact stimulation paradigm, and which amplitudes/frequencies were used (Figure 1D; reproduced later in this review on page 6). We note that the original manuscript pooled effects across all stimulation parameters, as we did not set out to test a parameter-specific effect. In response to reviewer feedback, we have now included a new analysis of frequency and amplitude effects (Supplemental Figure 2, reproduced here). Briefly, we observed that higher amplitude stimulation tended to evoke

greater increases in theta power, and marginally greater correlations with functional connectivity (i.e. TMI) – perhaps reflecting less noise in the measurement of theta power in remote sites. Using a repeated-measures ANOVA, we found no effect of frequency on evoked power or TMI, indicating that the correlation of functional connectivity with stimulation-induced theta power is not sensitive to the stimulation pulse frequency.

Supplemental Figure 2. Analysis of stimulation parameters on evoked power and theta modulation index (TMI). For each stimulation site, stimulation parameters were varied across amplitudes (three amplitudes, typically between 0.5-2 mA, 0.25 mA apart) and frequencies (10, 25, 50, 100, 200 Hz; see Methods for details). **(A)** The average theta (5-8 Hz) power evoked by stimulation (measured as the average pre-vs.-post T-statistic across the top 5 most strongly-connected electrodes to the stimulation target) is sensitive to stimulation amplitude, comparing the evoked power with the minimum delivered amplitude versus the maximum amplitude at each stimulation site (paired T-test, $P < 0.001$). The theta modulation index (TMI; see Figure 3 and Methods for details) is marginally sensitive to amplitude ($P < 0.1$). **(B)** Repeated measures ANOVA indicated no effect of stimulation frequency (measured at 10 Hz, 50 Hz, 200 Hz) on evoked power or TMI.

- Functional connectivity was defined as 5-13 Hz coherence, and authors state (Line 172) that “functional networks constructed from high frequencies (> 13 Hz) are not predictive of stimulation-induced theta activity”. While I agree that “functional connectivity” is a loose term, and any random measure could be chosen to denote such “connectivity”, I find the 5-13Hz coherence severely limiting. The same applies to gamma-gamma coherence. If we are to define functional connectivity as the intrinsic connectivity of two regions that are engaged together in a cognitive conditions, then we must show that the two “connected” regions have similar functionalities. Using this stringent criterion, and akin to fMRI BOLD functional connectivity maps in the same subjects, we have previously shown that the correlated infra-slow fluctuations of the high frequency broadband (high gamma, 70-150Hz) should be taken as the metric of such connectivity - I urge the authors to look at Foster et al (Neuron 2015 – see Fig 5E) and Kucyi et al (J Neurosci 2018) and summarized in Fox et (TICS 2018 – cited already in this manuscript). Coherence in 5-13Hz is a weak measure and I am not sure if it really reflects “functional” connectivity. I am familiar with the authors previous publications on coherent 5-13Hz (Nat Comm 2017), and in none of those publications the functional

connectivity in each subject was verified with anatomical and physiological selectivity of the responses during a cognitive task + fMRI resting state in et same subject. I do not think the authors should redo all their analyses, but they need to provide analyziz of <1Hz envelope of HFB instead of gamma-gamma coherence measures they have performed. This is especially important because authors identified connectivity on the basis of coherence in slow frequency and then showed that the slow frequency power only changes with stimulation in those areas that are “connected”. This seems like double dipping and circular argument even though the two samples are from entirely separate sessions (rest and recall).

The reviewer raises an excellent point, and we agree completely. While prior findings of cognitively-relevant, long-range low-frequency coherence (5-13 Hz) drove us to consider this as our primary connectivity metric, it has several limitations that the reviewer notes above. We agree that amplitude-envelope correlations of high-frequency broadband power offers another connectivity metric of potentially high functional relevance, as noted. In fact, we would like to take this suggestion one step further: Though rsfMRI was not collected in these subjects, atlas-based fMRI connectivity is available through the Human Connectome Project.

To consider other metrics of connectivity with established functional relevance, we have included analyses of stimulation-evoked power using two additional connectivity metrics: (1) <1 Hz fluctuations of HFB amplitude, as suggested, and (2) atlas-based rsfMRI connectivity. Both measures were less correlated with the location of evoked 5-8 Hz theta power than our original low-frequency coherence measure, and minimally predictive overall (main text Figure 5, reproduced here). We were surprised by this result, and hypothesized that HFB networks could be better predictive of power changes at higher frequencies (i.e. HFB connectivity may correlate with changes in HFB power).

Figure 5. Alternative measures of connectivity. (A) Example adjacency matrices for two subjects, reflecting functional connectivity from low-frequency (5-13 Hz) coherence, correlated high-frequency broadband amplitude (HFB; 50-200 Hz), and atlas-based resting state fMRI (rsfMRI). Matrices are organized as in Figure 2A. For reference, adjacency matrices of linearized Euclidean distance are shown at the far right. Colormap ranges are selected to visually emphasize network structure. (B) TMI is computed as in Figures 2-3, using adjacency matrices for each of the three measures. TMI was binned by distance from white matter, organized as in Figure 3B. In addition to significant TMI in and near white matter using 5-13 Hz coherence networks ($* P < 0.05$), we noted marginally significant TMI in white matter using atlas-based rsfMRI networks ($P < 0.1$). (C) Distribution of TMI for all stimulation electrodes regardless of distance from white matter. TMI is significantly greater at the group level for 5-13 Hz coherence ($P = 0.0002$). Note that the total count of stimulation electrodes is lower for rsfMRI connectivity ($n = 50$) analyses, due to subjects where atlas-based rsfMRI could not be extracted for a stimulation electrode; see Methods for details.

To test this, we further measured the correlations between connectivity and changes in the alpha/beta (10-25 Hz), gamma (30-50 Hz), and HFB (50-200 Hz) bands. First, we noted that stimulation in general tended to increase power at lower frequencies (<50 Hz) but decrease HFB power (Figure 6, reproduced here on page 13). Across all stimulation sites, the average

correlation of HFB connectivity with HFB power was not different from zero, but since these correlations are signed, we hypothesized that the average effect was dampened by individually strong positive and negative correlations. Indeed, we found 10 stimulation sites exhibiting significant positive or negative correlations between HFB power and HFB connectivity (7 negative), far exceeding the expected false positive rate (main text Figure 7, reproduced below). We concluded that the dynamics of stimulation-evoked power at higher frequencies is more complex, with a potential mixture of connectivity-related increases and decreases. These data suggest that HFB-based networks may be better at predicting decreases in HFB power, while 5-13 Hz networks excel at predicting increases in low-frequency power.

Figure 7. Power response at higher frequencies. (A) The average power modulation index – reflecting the degree to which functional connectivity predicts changes in spectral power – was computed for theta (5-8 Hz), alpha/beta (10-25 Hz), gamma (30-50 Hz), and HFA (50-200 Hz) bands for all 72 MTL stimulation sites. Functional connectivity was measured as 5-13 Hz coherence (left) and HFB amplitude correlation (right; see Methods for details). Across all stimulation sites, 5-13 Hz coherence significantly predicted changes in theta power, as demonstrated in Figure 2 (corrected $P < 0.05$). HFB amplitude correlations did not significantly predict power changes in any band. (B) To account for the fact that connectivity could predict decreases or increases in power, each stimulation site was classified by whether it’s functional connectivity profile significantly predicted decreases (blue) or increases (green) power (two-tailed $P < 0.05$). The count of significant stimulation sites are depicted as stacked bars; the expected false positive rate ($P = 0.05$) is indicated as a dashed line. Note that though HFB amplitude correlations do not significantly predict changes in HFA power on average (panel A), the number of stimulation sites where connectivity significantly predicts decreases in HFA power is significantly greater than chance (7 decreases, 3 increases; binomial test, corrected $P < 0.01$).

As a matter for further scientific discussion – but perhaps outside the scope of this manuscript – we appreciate that the Reviewer noted our findings of coherent 5-13 Hz activity in Solomon, et al. (2017), also in *Nature Communications*. It is true that we did not verify those connections with subject-specific physiological selective responses during a task (e.g. BOLD activation). However, we did show that low-frequency connectivity was significantly correlated with task-related increases in HFA (45-100 Hz) power, a band which substantially overlaps with the HFB measure shown to

correlate with fMRI BOLD responses (1). While this analysis was performed on the aggregated data set, and not across subjects, it is suggestive that low-frequency phase locking/coherence is related to known measures of functional activation. We agree completely that future work should seek to make these relationships more clear.

- While the authors have performed several control analyses to improve confidence that the effect was strongest for white matter electrodes, there remain at least two issues: Was one frequency of stimulation used at each electrode site (the protocol is not clear in the Methods)? If so, were the stimulation frequencies used at white matter sites comparable to those used at other sites? The authors state that “Inter-electrode coherences were computed for a series of 1-second windows (minimum of 10 windows per subject) extracted from the baseline period of a non-stimulation task, in which subjects wait passively before beginning a verbal free-recall task.” Does this mean that some baseline periods were only 10 seconds long whereas others were much longer? Was there a difference between patients who had white matter electrodes versus those who did not?

We apologize for the lack of clarity in our original methods section. For each stimulation site, whether in white or gray matter, we cycled through the same combination of frequency/amplitude stimulation parameters (10-200 Hz, 0.25-2.0 mA), and then aggregated across all parameters for the analyses presented in the original manuscript (see Figure 1D in the revised manuscript for a task overview, reproduced here). Additionally, the baseline coherence networks were computed by extracting 10 1-second windows from pre-task periods that usually occurred 24 times per recording session (coherences were then computed by averaging the cross-spectra across all 1-second windows). There is some variability in the number of pre-task windows available per subject, but each had at least 10 (corresponding to a minimum of 100 windows for estimating resting coherence networks). We recognize that the original methods were unclear on this point; the new Methods have been substantially revised for clarity.

Taken together, we do not believe there is any systematic difference in the treatment of white matter sites. As suggested by the original manuscript, even the distinction between near/far electrodes suggests a white matter dependence, as does the Pearson correlation between TMI and white matter distance (Supplemental Figure 3).

Figure 1D. Each stimulation site receives stimulation at three amplitudes and five pulse frequencies (50-200 Hz; see Methods for details). During each session, amplitudes are delivered in 60-trial blocks, within which 12 stimulations are delivered at each frequency. For the main results in this manuscript, effects are aggregated across all stimulation parameters; see Supplemental Figure 2 for analysis of stimulation amplitude and frequency.

- Could the authors provide more information/analysis of the anatomical locations of target electrodes? Did they include all white matter electrodes? Which regions comprised the “strong

connections” (Fig 4B)? I understand that this differs between patients, but a summary would be helpful.

All white matter electrodes in this MTL-stimulation dataset were included for analysis. The presence of white matter electrodes was confirmed by hand. (Though it is plausible that some “near” white matter electrodes are actually placed within.) The number of target electrodes in each MTL subregion is summarized in Supplementary Table 2 (the three most common targets are left CA1, right CA1, and left PRC).

The reviewer is correct that the “strong connections” differ between subjects. The most strongly-connected regions are typically cortical regions that are often covered by electrodes in this patient population. Among the most common are areas of the prefrontal cortex, lateral temporal cortex, and inferior parietal cortex (this summary has been added to the main text).

- Did the authors monitor for changes in behavior or subjective experience during stimulation? Please mention it briefly in the manuscript

We routinely (but informally) ask patients for subjective reports during stimulation sessions, though as subjects are asked to sit quietly during these sessions, we typically do not observe interesting changes in behavior. Subjects have not generally reported changes in subjective experience during these sessions.

- In the Discussion, the authors state that “no prior studies have addressed whether iEEG-based functional connectivity indicates anything about causal relationships in the brain, or whether it is merely a correlative measure.” This is disappointing. They should acknowledge previous work showing that fMRI functional connectivity is predictive of the effects of CCEPs (specially Keller et al 2011 PNAS; Shine et al 2017 J Neurosci).

We took care to not overstate our results, while at the same time acknowledging their novelty. We fully agree with the reviewer that prior studies have addressed the causal relevance of fMRI-based connectivity, and we aimed to fully acknowledge that (the Keller, et al. 2011 result is referenced in the original version one sentence prior to the quoted text, in addition to several others; we have added the Shine et al. 2017 reference in the revision). However, we do believe this manuscript is the first evidence that iEEG-based connectivity, specifically, is also predictive of stimulation-evoked effects. Such phase-based measures capture neural dynamics at substantially higher frequencies than the <1 Hz rhythms of fMRI (or HFB), and may yet be shown to carry information that is distinct from activity at slower timescales. We are happy to engage on this subject further, though we do think our original phrasing is accurate.

- Betzel, R. F. et al. reference details missing.

Reference has been updated:

Betzel, R. F. *et al.* Inter-regional ECoG correlations predicted by communication dynamics, geometry, and correlated gene expression. *arXiv* (2017). doi:1706.06088

- Line 84-87: These networks reflect correlated low-frequency activity between all possible pairs of electrodes in a subject. What does this mean?

We apologize if this sentence was unclear. Here, we are describing how we used 5-13 Hz coherence to estimate an adjacency matrix for each subject; that matrix summarizes the strength of connectivity between any two electrodes (e.g. “all possible pairs”) in a given subject. We hope this further detail makes our method clear!

- Precise number of stimulation targets: Why did we end up with 16 sites in the final analysis? How many subjects are these from? Can the authors be more clear (and transparent) about the selection of their stimulation target areas for the analysis. 40 sites were inside or near the white matter: But then the authors state (line 134) that only 16 stimulation sites exhibited significant TMI.

The 16 stimulation sites in the final network analysis were selected according to whether they exhibited an *individually* significant ($P < 0.05$) TMI, whereas there were 40 sites located in or near white matter. (Of the 16 significant TMI sites, 15 happened to be located in/near WM, though all 16 were used for these analyses.)

The reasoning for this distinction is as follows: Our first question (Figure 4A-B) was to ask whether connectivity-power correlations were driven by (1) increased power at strongly connected sites or (2) decreased power at weakly-connected sites. To answer this question, we only needed to consider stimulation sites for which there was a reliably positive correlation in the first place; therefore, we filtered by stimulation sites with significant theta power/functional connectivity correlations (i.e. TMI).

Our second question (Figure 4C-D) was to ask, “Do downstream power changes differ depending on whether a hub vs. non-hub was stimulated?” Though it would also be reasonable to ask this question among the 16 significant TMI sites, the range of hub scores in this relatively small subsample might not capture the true variability present in the data. Therefore, we considered the set of 40 sites within/near WM, which as a *group* exhibited significant TMI.

We have updated our Results text to make this distinction more clear (pg. 7-8).

- Paper refers to 33 electrodes near and 7 inside white matter and state that TMI for sites near or in white matter was significant. How much of this effect is due to the 7? What if you take out the 7? I personally think white matter targets should be analyzed completely separately. Stimulating inside the white matter in the proximity of MTL could be targeting connections coming from remote sites and stimulation could be inducing antidromic effects. Authors should please comment of antidromic effect versus prodromic. General reader would otherwise believe that the effects of stimulation always follow the direction of the axons. It could very well go antidromically as previously shown in the literature.

We appreciate the reviewer’s detailed feedback, and we agree that within-white matter sites deserve special consideration. The TMI (correlation between FC and theta power) is significant ($P < 0.05$) for sites “near” white matter alone ($n = 33$), as can be seen in Figure 3B. Additionally, at the group level, we found significantly positive connectivity-power correlations even if the 7 white matter contacts were not considered. We fully agree that stimulation could induce antidromic effects, and we regret not addressing this point in the original manuscript. We have incorporated this into the revised main text (see pg. 15).

- Line 122: It is very puzzling that theta activity induced by gray matter stimulation is uncorrelated

with functional connectivity to remote sites. But then the conclusions in the paper are stated as if the gray matter stimulations really mattered –though in lesser degrees. This needs to be discussed thoroughly.

We agree with the Reviewer on both points – it is puzzling that gray matter stimulation is less correlated with FC, we agree that the original manuscript is at times contradictory on this point. Throughout the text, we have made revisions to clarify our thinking, as follows:

We do not want readers to over-interpret the low gray matter TMI reported in Figure 3B. Though not significant, the effect in gray matter is above zero, and certain gray matter sites exhibit individually high/significant TMI. In fact, we would not be surprised if the gray matter effect becomes significant as more data is collected. There likely is some correlation between gray matter stimulation and FC, but it appears to be numerically and statistically lower than the correlation observed at sites closer to white matter – if stimulation is principally exerting remote effects via direct engagement of white matter tracts, this relationship makes sense. Indeed, our “near white matter” category is, in reality, still referring to an electrode likely placed in gray matter. Furthermore, as noted in the Discussion, gray matter stimulation within the MTL may exhibit different dynamics than gray matter stimulation on the cortical surface, which was not assessed here.

- Line 130: Please remove the mention of DIRECTIONALITY. functional connectivity does not necessarily denote directionality.

We apologize for the use of overloaded terminology. In this case, we were referring to the “directionality” of theta power changes, i.e. whether power increases or decreased. The sentence has been reworded for clarity. (As mentioned in a response to an earlier request, we have also commented on the inability of our functional networks to predict prodromic/antidromic propagation.)

Similarly, line 201 states that “results generally support the idea that functional connectivity indicates causality”. This is very controversial and depends on how one defines causality. Please explain this and or take away the notion of causality. It is really not that relevant to the findings of the study anyway.

Our purpose with this terminology was to underscore how use of intracranial stimulation in a substantial dataset (26 subjects) could reveal neural dynamics unapproachable with traditional methods. Our study demonstrated that low-frequency coherence of iEEG correlates with the effect of intracranial stimulation. It is true that “causality” could be interpreted in several ways – the effects of stimulation and coherence may both relate to a third factor (such as the white matter scaffold of the brain) that truly “causes” an observed effect. Indeed, there are likely many cases where low-frequency coherence is low, but a causal effect of stimulation can still be observed. In recognizing this complexity, we have reworded the sentence in question as follows:

“Our results generally support the idea that functional connectivity indicates causal relations in the brain; when stimulation occurs in or near white matter, we could predict where power changes would occur based on distance-independent measures of low-frequency functional connections.”

- Line 147: Regions with strong connectivity to the rest of the brain, are generally less capable of

modulating the brain's overall state versus non-hubs. This seems not right. How was the "hob-ness" measured? Can the authors please explain?

We fully agree that the finding of decreased power changes downstream of a hub is a nonintuitive finding, and we have substantially revised the Discussion to address this point, including a softening of the language surrounding this result (see pgs. 14-15). The revised Discussion now raises several alternative explanations for this result, including the possibility that hub stimulation does evoke significant widespread change, but outside the theta range. Additionally, we have attempted to clarify how this result aligns with predictions of network control theory.

In this study, "hubness" was measured as the average strength of connectivity between a stimulation and all other electrodes in the brain (i.e. the node strength (2)). In the analysis in Figure 4, hubs scores are divided into terciles, and the magnitude of downstream theta power change is computed for the upper and lower terciles. In this way, we found that downstream power changes are higher for stimulation sites with lower hub scores (i.e. regions that are sparsely functionally connected).

Briefly, the revised Discussion now describes how network control theory suggests the hubness of a node relates to its ability to alter the overall state of a whole-brain network (Gu, et al. 2015; Muldoon, et al. 2016; Medaglia, et al. 2016) – strong hubs may more easily influence the state of the brain overall, but stimulation weak hub nodes may help move the brain to more difficult-to-reach states. It is therefore of great interest to ask, empirically, whether the effects of stimulation at hubs vs. non-hubs differ. However, we acknowledge that the mapping between "brain states" and theta power is not clear; furthermore, the partial-brain coverage in each subject and the focus on MTL stimulation in this dataset limit our ability to strongly generalize this finding. Therefore, we offer this analysis as an intriguing initial data point in what will hopefully be a long line of empirical research testing theories of network control. See the revised Discussion for further detail.

- Line 220 says: the findings "bridge prior studies of stimulation and behavior with underlying neural mechanisms". I think the authors should water down this statement. I do not think the current study reveal a mechanistic explanation for how stimulation and behavior are related.

**We appreciate the feedback. The sentence has been rewritten as follows:
"However, with an eye towards the eventual therapeutic use of stimulation, the results here begin to bridge prior studies of stimulation and behavior with underlying neural mechanisms."**

Overall, I congratulate the team for producing this excellent piece and I hope that my comments could improve its quality even further.

Reviewer #2 (Remarks to the Author):

The authors present a well written manuscript describing low frequency (theta and alpha) activity in the mesial temporal lobe and how the propagation of these events relate to functional connectivity methods from a series of 26 neurosurgical patients. While the topic is timely, the analyses appear

well motivated and the manuscript is well written, a number of concerns dampen my enthusiasm for publication in Nature Communications as outlined below:

1. My main concern regards the relationship between electrical stimulation and induction of the theta rhythm. While Figure 1 shows an example of pre stim and post-stim theta, it is not clear whether this represents a broad band increase in power (as would occur in the case of a stimulation-induced afterdischarge) or the authors hypothesis that this phenomenon is selective to lower frequencies. Without a spectrogram depiction both for the individual case as well as some summary of the entire data set (as attempted in figure 1E), it is not possible to determine the relationship between specific frequency bands and any effects demonstrated.

We fully agree that the frequency-specificity of stimulation effects is an interesting and important point. Our original intent was to specifically test theta-band power, given an especially rich literature showing the cognitive relevance of theta oscillations. This theoretically-motivated choice served to focus our hypothesis and reduce the number of tests to statistically assess other frequency bands.

However, as the Reviewer notes, it is pertinent if changes in power are broadband, or specific to theta. With this in mind, we have included new analyses that measures the evoked power and modulation index in alpha/beta band, low gamma, and HFB (50-200 Hz) bands. First, we measured the stimulation-induced change in spectral power in each band, and averaged the effects across all recording electrodes for each stimulation target. We took this as a measure of whole-brain induced power. As shown in Figure 6C, whole-brain power increases are observed in the theta, alpha/beta, and gamma bands, while power decreases are observed in HFB. (An example subject is shown in Figure 6A-B.)

Furthermore, we examined whether specific electrodes which exhibited a significant theta increase also showed a concomitant increase in power at higher frequencies. To do this, we averaged the induced power across only the recording electrodes which exhibited a statistically significant stimulation-induced elevation in theta power. At these electrodes, significant power increases were also observed in the alpha/beta and gamma bands, but with lower magnitudes than the theta increase, and no increase was observed in HFB (Figure 6D). Therefore, these electrodes exhibit a selective theta response, with some bleed through to the alpha/beta and low gamma bands.

The pertinent figure has been added to the main text (Figure 6).

Figure 6. Stimulation-induced power across frequency bands. (A) Trial-averaged stimulation-induced power spectrogram for an example electrode from a single subject (stimulation in left MTL white matter, recording electrode in left inferior parietal cortex). (B) Post-minus-pre stimulation difference in power from the electrode in (A). (C) Whole-brain stimulation-induced power was measured by computing a T-statistic on the pre- vs. post-stimulation spectral power at each electrode in a subject, and then averaging across electrodes to get an estimate of whole-brain change in power. On average, stimulation tended to elevate whole-brain power in the theta (5-8 Hz), alpha/beta (10-25 Hz), and gamma (30-50 Hz) bands. Stimulation tended to decrease power in the HFB range (50-200 Hz). (D) For each subject/stimulation site, electrodes were classified by whether they exhibited a significant ($T > 2$) change in theta power induced by stimulation (at least 1 theta-responsive electrode was found for 47 of the 72 stimulation sites). The stimulation-induced change at higher frequencies was computed for this subset of electrodes, to determine whether the power response was specific to theta. Across all subjects/stimulation sites, increased power was observed in the alpha/beta and gamma range at theta-responsive electrodes, but no effect was observed in the HFB range. The bar for induced theta power is delineated by a dashed line and shown as a reference only, since theta power was the basis for selecting these electrodes for further analysis.

2. They used a range of electrical stimulation from 10-200 Hz, but do not detail what specific stimulation frequency results in the increase in theta power. If the premise here is that electrical stimulation induces theta in the first place, then it would be quite valuable to know exactly how to increase theta power in the hippocampus so results may be replicated and the field could advance. A particular concern relates to the fact that because of this, the analyses may be subject to cherry-picking the most consistent results from one particular stimulation frequency (out of the 5 different ones used). This concern further makes figure 1E difficult to interpret.

The reviewer raises a good concern – in fact, our original approach was to aggregate the effects of stimulation across all frequencies (and amplitudes), to explicitly avoid cherry-picking a

spuriously efficacious set of stimulation parameters. In doing so, we make the claim that stimulation, in general, evokes connectivity-related increases in low-frequency power. We did not make a claim about what frequency of stimulation achieves this.

However, we agree that the evoked theta effect may be specific to particular stimulation frequencies. To analyze this for the revised manuscript, we measured TMI (the correlation between evoked theta and functional connectivity) and raw evoked power as a function of stimulation frequency. We found no effect of stimulation frequency on raw evoked power or TMI (see Supplemental Figure 2B). This suggests that theta power increases were driven by stimulation in general, not a specific pulse frequency. For more detail and the reproduced figure, see response to Reviewer 1 (pg. 3).

Based upon these two major concerns, the main findings here are that signals (presumably causal to stimulation) propagate to areas of connectivity. While this is demonstrated here, the findings are largely predictable. The efficacy of white matter stimulation and the differential effects of hub/non-hub and strong/weakly connected areas is intriguing, but would benefit from more detailed analyses considering correlation of DTI and alternate resting connectivity measures.

We greatly appreciate the Reviewer's thoughtful feedback. In addition to addressing the two chief concerns above with new analyses, we have also related power modulation to two new functional connectivity metrics: (1) Amplitude envelope correlation of HFB, and (2) atlas-based rsfMRI connectivity. We have performed extensive new analyses to show that our original measure, low-frequency coherence, appears to better predict increases in theta power than HFB or rsfMRI networks. However, HFB networks do appear to capture the stimulation-related power decreases in the HFB band itself. For more details and a new main text figure, please see the response to Reviewer 1 on pages 5-6.

Reviewer #3 (Remarks to the Author):

In this study, Solomon et al apply electrical stimulation to the medial temporal lobe in 26 neurosurgical patients to answer the question of how external perturbation modulates brain networks. They find networks of theta coherence predicted stimulation-evoked changes in theta power, but only when stimulated electrodes were in white matter. Furthermore, they find that when network hubs were stimulated, it evoked less theta changes than when non-hubs were stimulated. This question is important to better understand the mechanism underlying brain stimulation treatments, and its potential to optimize and personalize these treatments. The study is potentially highly impactful, but in my opinion a major revision is necessary before publication. My questions / comments are below:

We appreciate the reviewer's helpful and detailed feedback! In response, we have made substantial revisions to the manuscript and included new analyses. Each specific concern is addressed below.

1] I found the methods and amount of terms in the study to be confusing, and takes the reader away from the important findings in here. I would recommend a careful review to see if any abbreviations can be removed or simplified. TMI was difficult to understand. Additionally, for example, in figure 2, terms like BP, LFA, Logit are not clear from the figure or the legend.

We apologize for the confusing acronyms. Throughout the manuscript, we have simplified terms and removed acronyms where possible. Additionally, we have replaced the “TMI” acronym with “network-mediated activation,” or NMA, which we believed is a clearer description of the statistic (within this document, we have retained use of TMI to be consistent with reviewer comments).

2] I would recommend adding a discussion about a recent publication by Keller et al., JNeuro, 2018 as it is relevant to this work. A few sentences in the discussion would be helpful to compare this current study with the recent publication, and how it either complements or contrasts the findings and methods on how to explore/residualize distance and the relationship between functional connectivity and stimulation-induced changes observed.

The Keller, et al. 2018 study came to our awareness after our initial submission of this manuscript. We agree that it is highly relevant, and we have now cited it in the Introduction and Discussion. Furthermore, we discuss the present manuscript in relation to the Keller 2018 study in a new Discussion paragraph (see main text pg. 15). The pertinent text has been copied below:

“The findings from this study could be extended in several ways. A recent study by Keller, et al. (2018) asked whether a multivariate model could predict how direct brain stimulation alters remote cortical excitability(3). A similar approach could be adopted with these data, wherein multimodal measures of connectivity – e.g. coherence, HFB-envelope, and white matter proximity – could be used to predict the stimulation response across locations and frequency ranges. Such an approach could reveal relationships that were obscured by the univariate methods in this manuscript; gray matter targets, for instance, may induce widespread, connectivity-related changes in specific frequencies that are predictable by a weighted combination of functional networks.”

3] In general, I had a very hard time determining what stimulation frequencies were used. In the methods, it states that a variety of amplitude, frequency, and duration was used, in an event-related design. This should be very clear to the reader which stimulation parameters were used for which analyses, and be clearly reflected in the figures and text. Maybe I missed it, but I was under the understanding that only theta frequency stimulation was examined, which would make sense that theta stimulation elicits theta-specific changes post-stimulation. More interesting would be to show that for other stimulation frequencies, either only theta modulation was observed, or the modulation observed was dependent on the stimulation frequency. This is an important question that should be addressed in this study and not a future study, especially because the data has been collected and strong statements are made about theta specifically, but may instead be purely dependent on the stimulation frequency applied. My apologies if this was clear but I just did not fully understand it.

We agree with the Reviewer; our original description of stimulation frequencies was unclear. All analyses in the original manuscript used stimulation trials aggregated across all frequency and amplitude parameters; in other words, we reported effects of 500ms stimulation in general, regardless of frequency/amplitude. This was intentional, as we avoided additional tests and the possibility of cherry-picking a spuriously efficacious combination of stimulation parameters. To clarify this, we have added a new experiment schematic in Figure 1D, reproduced earlier in this document in response to Reviewer 1 (page 7).

However, as other Reviewers have mentioned as well, changes in theta power could be specific to a certain stimulation frequency. To test the possibility, we have included a new analysis in the revised manuscript that examines theta modulation as a function of stimulation frequency (we considered 10 Hz, 50 Hz, and 200 Hz stimulation; see Supplemental Figure 2, also reproduced earlier in this document on page 3). As described in the response to Reviewer 1, we found that evoked power and power-connectivity correlations were insensitive to stimulation frequency (though evoked power was higher for higher-amplitude stimulation).

4] In many ECoG studies, (like this one) often electrodes across subjects need to be grouped together in order to achieve sufficient power for an analysis. I would be curious how many subjects the 7 ‘within WM’ electrodes came from? It would be relevant if it is from 7 patients or from 1.

This is an excellent point, we and we apologize for not making the distribution of data clear. The 7 within-WM points came from 6 subjects; one subject contributed two separate stimulation sites. The locations of these sites are displayed on MRI in Supplemental Figure 1.

5] Regarding the hub analysis, in the abstract you state, ‘these power changes aligned with control-theoretic predictions of how exogenous stimulation flows through complex networks’ and later ‘while counterintuitive, this result is in line with the prediction of network control theory.’ I think this finding warrants further discussion. Finding that stimulation of a hub region elicits less modulation than a non-hub region is the most interesting finding in this paper, in my opinion. Most people would have thought a strong hub region would have more ability to modulate regions within the network. If true, this has implications for the fields of non-invasive and invasive brain stimulation. However, the authors have minimal citations regarding this counterintuitive finding being in line with network control theory, and instead state it as if it is a given, which my guess is most readers would think otherwise. A longer discussion on this point is warranted. If space is an issue, I found paragraph 3 in the discussion to be slightly redundant.

We fully agree that the finding of decreased power changes downstream of a hub is a nonintuitive finding, and we have substantially revised the Discussion to address this point, including a softening of the language surrounding this result (see pg. 14). The revised Discussion now raises several alternative explanations for this result, including the possibility that hub stimulation does evoke significant widespread change, but outside the theta range. Additionally, we have attempted to clarify how this result aligns with predictions of network control theory:

Briefly, the revised Discussion now describes how network control theory suggests the hubness of a node relates to its ability to alter the overall state of a whole-brain network (Gu, et al. 2015; Muldoon, et al. 2016; Medaglia, et al. 2016) – strong hubs may more easily influence the state of the brain overall, but stimulation weak hub nodes may help move the brain to more difficult-to-reach states. It is therefore of great interest to ask, empirically, whether the effects of stimulation at hubs vs. non-hubs differ. However, we acknowledge that the mapping between “brain states” and theta power is not clear; furthermore, the partial-brain coverage in each subject and the focus on MTL stimulation in this dataset limit our ability to strongly generalize this finding. Therefore, we offer this analysis as an intriguing initial data point in what will hopefully be a long line of empirical research testing theories of network control. See the revised Discussion (pg. 14-15) for further detail.

6] Pre-stimulation theta calculation. If my understanding is correct, the authors applied repetitive

stimulation trains in an event related design every 3 seconds, and used the 900ms before the stimulation train for the baseline period. Although likely enough time, it would be worthwhile to show that the train before did not influence the next train's pre-stimulation baseline period. Showing a control analysis of pre/post compared to pre/post2s (1 or 2s after the train) would be helpful and clarify this.

The Reviewer's understanding is correct. In response to this critique, we are including a new "early baseline" control analysis here, as suggested. The post-stim theta power is compared to pre-stim power starting 2 seconds prior to the stimulation train, instead of 1 second as in the manuscript. Though slightly reduced, the main effect remains significant, as does the relationship with white matter.

Analysis of earlier pre-stimulation baseline. To account for the possibility that post-stimulation effects from one event persist until the pre-stimulation baseline (-950 ms to -50 ms) of the next event, we re-analyzed our key result using a pre-stimulation baseline shifted earlier by 1 second (-1950 ms to -1050 ms). Results were qualitatively consistent with the original baseline periods; greater effects were observed in stimulation targets placed closer to white matter, and the effect across all stimulation targets remained significant ($T(71) = 3.2, P < 0.01$).

7] Figure 2B - different colors for SOZ and artifact so there is a clearer contrast between regions. Same for D - black lines would be clearer, as you're already using red in the figure to denote the T-stat strength.

The figure has been updated according to these suggestions.

8] Stimulation artifact. The authors state there is an artifact lasting for up to 350ms after the last stimulation pulse. However, I wonder if this is neural vs non-neural. Can the authors show what this artifact looks like? 28% of channels were removed because of this procedure, and if neural in origin they may be removing nearly a third of the most strongly connected regions.

Please see the below figure, also included as Supplemental Figure 4, for a visualization of the typical post-stimulation artifact. We fully agree that it is pertinent to ask whether the artifact may be neural in origin. However, we believe this highly consistent slow decay is nonphysiologic and likely reflects a hardware-level response to repeated stimulation pulses. (We further confirmed this with a bench test of stimulation in a saline tank, demonstrating the same slow decay profile.)

Supplemental Figure 4. Depiction of post-stimulation artifact. A subset of channels (average of 28% per subject) exhibited a non-physiologic post-stimulation artifact, characterized by a slowly decaying voltage offset immediately after the last stimulation pulse. A typical example of this artifact for one stimulation event, is shown on the left, and 10 representative examples are shown on the right, demonstrating their consistency across events. The red bar indicates the 350 ms post-stimulation period used to assess a channel for rejection (see Methods). Blue dashed lines indicate the 500 ms stimulation interval.

9] Regression procedure. I understand and applaud the authors for regressing out distance as it is clearly an important and often overlooked factor. Some analysis or citations of how distance is often correlated with strength of response would be helpful. On that note, it may not be necessary, but should at least be commented on why a multivariate regression model wasn't used to predict regions of modulation, using features including distance, coherence, and theta power, similar to the recent JNeuro paper by Keller et al. This way one can see the amt of theta power explained by anatomical distance vs coherence. Other related questions include compared to absolute distance measure, how well does functional connectivity predict TMI? (Provide the R^2 for both). And is distance colinear with functional connectivity? If so, then the resulting linear regression will be unstable and this should be addressed.

We appreciate the Reviewer's feedback on this important point. While it would be possible to separate out the contribution of distance and coherence as explanatory factors of theta modulation (similar to the Keller, et al. 2018 study), we chose here to try to address our hypothesis as simply as possible: Does coherence, independent of distance, predict stimulation-evoked change? We agree that a multivariate approach could yield more insight into the relationship between coherence/distance and evoked power changes. However, as our goal here was to quantify the extent to which functional connectivity predicts stimulation effects – and not to fully explain stimulation effects – we feel our current approach is reasonable and more directly suited to the hypothesis at hand. We have expanded our discussion of this point (see pg. 15) and included a consideration of a future multivariate approach, as suggested.

To address the Reviewer's question regarding collinearity, we examined the correlation between iEEG coherence and distance measures. Pearson/Spearman correlation coefficients are, on average, approximately 0.5, while the average variance inflation factor (VIF) is 1.49. This is well below the rule-of-thumb cutoff of 5, indicating collinearity is not a significant concern. Furthermore, we use a permutation test to assess the significance (e.g. z-score) of the beta-weight on functional connectivity, minimizing the risk that our data violate model assumptions.

10] In general, I found some of the discussion text to be unnecessarily strong, especially around the

hub finding. A softer discussion and one that offers other alternative explanations as to why the counterintuitive finding was observed, would be interesting and helpful for future studies.

In response feedback from the Reviewers, we have substantially revised the discussion -- including a softening of the language -- and specifically added additional detail to the counterintuitive hub finding. We have reproduced the pertinent section here (pg. 14-15):

“In this study, we also assessed the relationship between stimulation and the network topology of a targeted region. Specifically, we asked whether the downstream effects of stimulation differed between hubs and non-hubs, reflecting regions that are richly or sparsely connected. Counterintuitively, we found that stimulation of non-hubs yielded greater increases in theta power at downstream sites. It is possible that (1) hub stimulation does result in greater distributed power changes, but outside the theta band, or (2) hub stimulation results in a dispersal effect, causing widespread change but limiting the magnitude of the effect at any single downstream site. Principles of network control theory postulate that stimulation of sparsely connected regions can be efficacious for moving the brain to “difficult-to-reach” states, or states that require significant cognitive effort to achieve (4–6). However, the mapping between spectral power and “brain states” in a cognitive sense remains unclear; further empirical and theoretical work should aim to clarify how control theoretic predictions can be tested with common intracranial techniques.”

11] I found the lack of animal literature in the paper to be striking. Is there really minimal studies examining the relationship between functional connectivity and causal connectivity? If so, it should be more clearly stated. If not, citations should be added.

We regret not including more references to prior research on this subject in animal models. While we had originally cited the study by Logothetis, et al. in 2010 which investigated stimulation/connectivity in primates, we have added several additional, highly relevant citations in the revised Introduction and Discussion (macaque and rodent):

- Tehovnik, E. J. Electrical stimulation of neural tissue to evoke behavioral responses. *J. Neurosci. Methods* **65**, 1–17 (1996).
- Yeomans, J. Electrically evoked behaviors: axons and synapses mapped with collision tests. *Behav. Brain Res.* **67**, 121–132 (1995).
- Tolias, A. S. *et al.* Mapping Cortical Activity Elicited with Electrical Microstimulation Using fMRI in the Macaque. *Neuron* **48**, 901–911 (2005).
- Ranck, J. B. Which elements are excited in electrical stimulation of mammalian central nervous system: a review. *Brain Res.* **98**, 417–40 (1975).

12] The paper would benefit from a discussion about the type of stimulation applied (examining short-term plasticity), and how it contrasts with single pulse CCEP studies (probing and very temporary changes) to single train application to longer-term changes as in Keller et al paper and as is applied in TMS treatment for depression.

The Reviewer raises an important point that was not addressed in the original manuscript. The method of stimulation has a potentially profound effect on downstream changes in neural activity, and many prior intracranial studies have leveraged CCEPs instead of sustained oscillatory activity induced by stimulation pulse trains (something we believe makes our work

novel). In the revised Discussion, we have incorporated a new discussion of how this study relates to prior Keller studies, noting differences in the stimulation paradigm. Furthermore, in the Discussion we compare and contrast the current study to alternate stimulation protocols (pages 15-16).

13] Is there a possibility the theta power increase simply reflects the evoked potential of the last pulse? It may be worth exploring what happens when you remove the first 500+ms of the last pulse in the train, as the evoked potential can certainly last up to 1s, and the slow wave can be in the range of theta.

To address this possibility, we performed a new analysis as the Reviewer suggested: We only computed theta power for a “late” 500 ms window, leaving an unanalyzed 500 ms gap after stimulation. While the main TMI effect is reduced in magnitude, it is qualitatively consistent with the findings in the main paper; functional connectivity can predict remote increases in theta power, increasing with proximity to white matter. These results suggest that stimulation-induced theta activity is indeed sustained for at least 1 second. (We also note that removal of the first 500 ms after stimulation, while controlling for the influence of an evoked potential, also removes potentially genuine theta oscillations that occur in that timeframe.)

Analysis of “late” 500 ms post-stimulation window. To control for the possibility that a non-oscillatory evoked potential contributes to measures of theta power in the immediate post-stimulation period, we reassessed TMI in a 500 ms window spaced 500 ms from the offset of stimulation. Though the magnitude of the effect was reduced from the full 900 ms window analysis, results were qualitatively consistent with our main results. “Near white matter” contacts showed a marginally significant TMI ($P < 0.1$).

14] What does the spectrum of power look like after the train? Is there a selective increase in theta power or do all other frequency band also see increased power? i.e. how selective is this theta power increase?

This is an excellent question, and we sought to answer it with a detailed new analysis included in the revised manuscript. We have addressed it in detail in a response to Reviewer 2 (pg. 12), but summarize here:

For each stimulation site we asked whether at least 1 recording contact showed a significant ($T > 2$) evoked theta response (a “theta responsive electrode”). We found at least one theta responsive electrode for 47 of the 72 stimulation sites. Next, among those theta responsive

electrodes, we asked whether there was a significant modulation in the alpha/beta, low gamma, and HFB bands (Figure 6D). While significant power increases were observed in alpha/beta and gamma, they were weaker than the evoked theta response (no significant elevation in HFB was observed). This indicates that, among electrodes that show a strong theta response, the effect is reasonably specific to that frequency band. An example spectrogram for a single electrode is depicted in Figure 6A-B, which demonstrates a maximal response in the theta range.

More broadly, we also asked how stimulation affects spectral power more generally. Averaged across all recording electrodes, we observed stimulation-induced increases in theta, alpha/beta, and gamma power, but decreases in HFB (Figure 6C). Both of these findings are described in the revised main text Results section and a new figure.

References

1. Fox KCR, Foster BL, Kucyi A, Daitch AL, Parvizi J (2018) Intracranial Electrophysiology of the Human Default Network. *Trends Cogn Sci* 22(4):307–324.
2. Rubinov M, Sporns O (2010) Complex network measures of brain connectivity: Uses and interpretations. *Neuroimage* 52(3):1059–1069.
3. Keller CJ, et al. (2018) Induction and quantification of excitability changes in human cortical networks. *J Neurosci*:1088–17.
4. Muldoon SF, et al. (2016) Stimulation-Based Control of Dynamic Brain Networks. *PLOS Comput Biol* 12(9):e1005076.
5. Gu S, et al. (2015) Controllability of structural brain networks. *Nat Commun* 6:8414.
6. Medaglia JD, Pasqualetti F, Hamilton RH, Thompson-Schill SL, Bassett DS (2016) *Brain and Cognitive Reserve: Translation via Network Control Theory*.

REVIEWERS' COMMENTS:

Reviewer #1 (Remarks to the Author):

I think the quality of the paper has increased substantially. I thank the authors for their careful revision.

Reviewer #2 (Remarks to the Author):

The authors have addressed my main concerns and have done a thorough job of resolving them. I have no further issues.

Reviewer #3 (Remarks to the Author):

I thought the reviewers addressed all of my concerns and I think it is of scientific importance to be published in Nature Communications.